# The differentiation of myeloid progenitors is effected by cascading waves of coordinated gene expression that remodel cellular physiology in a characteristic sequence

Andrea Repele[1☉], Joanna Handzlik[1☉], Nimasha Samarawickrama[2], Trevor Long[1], Sunil Nooti[1], Veena Potluri[3], Yen Lee Loh[2], Manu[1]*

1 Department of Biology, University of North Dakota, Grand Forks, North Dakota, United States of America, 2 Department of Physics, University of North Dakota, Grand Forks, North Dakota, United States of America, 3 Pathology, Altru Health System, Grand Forks, North Dakota, United States of America

☉ These authors contributed equally to this work.
* manu.manu@und.edu

## Abstract

The differentiation of hematopoietic progenitors into specialized types requires the transmittal of information from a few external or internal regulators to the thousands of genes that produce a cell type's characteristic phenotypes. While the main signaling pathways, transcription factors, and the genes eliciting the terminal phenotypes are known, how information flows from a few regulators to thousands of genes to change the state of the cell remains to be fleshed out. To profile this information transfer process, we sampled the differentiation of the PUER myeloid cell line into macrophages and neutrophils at 27 time points over seven days. There is extensive transient regulation; the number of transcripts modulated in time is twice the number differentially expressed between endpoints. Differentiation is marked by two sharp transitions, at ~8h and ~80h, when transcriptomic state changes suddenly. We utilized non-negative matrix factorization to identify *behaviors*, characteristic temporal patterns of gene expression, and to classify transcripts by behavior. Only 10 distinct behaviors are sufficient to recapitulate the expression of ~36,000 transcripts with high fidelity. Gene expression in most of the behaviors occurs in pulses of varying initiation times and durations. This implies that information transfer during differentiation occurs in cascading waves of gene expression culminating in the permanent turning on of certain genes after ~80h. Each behavior is enriched in specific biological processes, so that physiological remodeling proceeds in a characteristic order—signal transduction, translation and mRNA processing, metabolism, and, ultimately, myeloid phenotypic processes. The sharp transition at 8h corresponds to the completion of transcriptional and translational remodeling and the initiation of metabolic remodeling; the one at 80h corresponds to the elicitation of myeloid phenotypes. Our analysis shows that differentiation relies upon a series of transient, rapid, and complex gene

**Data availability statement:** The RNA-Seq and CUT&RUN data are available at National Center for Biotechnology Information's Gene Expression Omnibus at accession numbers GSE326003 and GSE325971 respectively. NMF code is available on Github at https://github.com/mlekkha/NMF.git.

**Funding:** This work was supported by the National Science Foundation (https://www.nsf.gov) [1942471 to M.]. The funder did not play any role in the study design, data collection and analysis, decision to publish, or preparation of the manuscript.

**Competing interests:** The authors have declared that no competing interests exist.

regulatory events and highlights the importance of profiling it at a high temporal resolution.

## Author summary

The maturation of hematopoietic progenitor cells into differentiated cell types occurs over a period of about a week. This process requires the progenitors to respond to external signals by changing the expression of thousands of genes to elicit the required phenotypes. We profiled how information is transferred from a few upstream regulators to thousands of genes by measuring genome-wide gene expression at high temporal resolution during white-blood cell differentiation. We show that the information transfer occurs in cascading waves, some as short as 8 hours and others lasting for 3 days, in which thousands of genes change expression coordinately. The physiological processes remodeled in each successive wave follow a characteristic order, starting with signal transduction pathways, followed by translation and mRNA processing, then metabolism, and culminating in the production of innate immunity phenotypes. Maturation is also punctuated with two sharp transitions, when genome-wide expression changes suddenly, associated with the initiation of metabolic remodeling and the production of terminal phenotypes. Our analysis shows that a complete description of differentiation requires the characterization of transient changes and not just those observable at the endpoints.

## Introduction

Macrophages and neutrophils are developmentally closely related cell types that share a common progenitor, the granulocyte-monocyte progenitor (GMP) [1,2]. The choice of developmental fate, macrophage or neutrophil, is understood to involve well-known hematopoietic transcription factors (TFs) that regulate each other in the manner of a bistable switch [3,4]. This decision is thought to depend on the ratio of the expression of the TF PU.1, known to be essential for the development of all white blood cell lineages [5], and C/EBP$\alpha$, whose expression is necessary for neutrophil development [6]. A high PU.1-C/EBP$\alpha$ ratio was shown to promote the macrophage fate, while a low ratio favored neutrophil development [7]. The actual bistable switch effecting the decision is believed to be downstream of PU.1 and C/EBP$\alpha$. One proposal involves mutual repression between the TFs Egr1/Egr2/Nab2 and Gfi1, promoting the macrophage and neutrophil fates respectively [3]. Another bistable switch may operate between Irf8 and Gfi1 [4] either in exclusion of the Egr1/Egr2/Nab2-Gfi1 switch or in complementarity of it.

 While we have some insight into the key regulators that bias the initial conditions towards one fate or the other, we know relatively little about how these initial biases are converted into the ultimate phenotypic differences observable between distinct

cell types. Comparisons of hematopoietic cell types typically reveals hundreds to thousands of differentially expressed genes [8]. In *in vitro* differentiation [7,9] and transdifferentiation [10] experiments, the initial bias could be introduced by modulating the expression of a single TF. This raises several outstanding questions about how information flows from a few upstream regulators to thousands of genes to remodel the transcriptome of a terminally differentiated cell and elicit its observable phenotypes. Are the genes that produce the observable physiological phenotypes induced early or late during differentiation? When are the determinants of the identity of the cell, such as TFs and signal transduction factors, induced and, conversely, when are the determinants of the progenitor and alternative lineages downregulated? Do any genes change expression transiently and revert in expression to their state in progenitor cells? On what time scales do these changes occur? To summarize, relatively little is known about the timing, sequence, and scale of genome-wide gene expression modulation during macrophage-neutrophil differentiation [11,12].

In order to profile the timing and sequence of gene expression changes as they snowball from a few genes to the scale of the genome, we acquired a high resolution RNA-Seq time series dataset of *in vitro* macrophage-neutrophil differentiation of PU.1 estrogen receptor (PUER) cells, an important model of myeloid development [7,13,14] (Fig 1A). PUER cells are an IL3 dependent hematopoietic progenitor cell line derived from the fetal liver of PU.1$^{-/-}$ mice in which PU.1 has been reintroduced after fusion to the ligand binding domain of the estrogen receptor [15]. In the absence of 4-hydroxy-tamoxifen (OHT), the PUER fusion protein is inactive so that uninduced PUER cells have a PU.1$^{-/-}$ phenotype and can be maintained indefinitely as bipotential progenitors by culture in medium containing IL3. OHT activates the PUER fusion protein by binding to the estrogen receptor domain, restoring the function of the PU.1 TF and relieving the block on the progenitors to induce differentiation. PUER cells can be differentiated into macrophages by OHT treatment over a period of 168 hours and into neutrophils by substituting GCSF for IL3, culturing in GCSF medium for 48 hours, and subsequently treating with OHT for 168 hours. Fate transformation in PUER cells therefore, is initiated by just two inputs, the TF PU.1 and the cytokine GCSF, making the system a good candidate for observing the information transfer process in a bifurcating cell-fate decision.

Our analysis of the RNA-Seq data shows that genes are expressed in diverse and complex temporal patterns throughout the course of differentiation. Differential expression analysis between the end points, while identifying thousands of genes that change in expression, underestimates the number of genes modulated in expression during differentiation since a large number of genes change expression only transiently. To overcome this limitation of pairwise comparisons, we utilized non-negative matrix factorization (NMF) to identify characteristic temporal patterns of gene expression over the entire duration of differentiation, called "behaviors", and to cluster the genes by similarity to the behaviors. We found that only ten behaviors were sufficient to account for the temporal expression patterns of ~36,000 transcripts and that each behavior was exhibited by hundreds to thousands of genes. Many of the behaviors were pulsatile so that expression was temporally restricted to periods ranging from 8 hours to 3 days. Gene ontology (GO) analysis showed that each behavior is enriched in specific physiological processes so that information is transferred in a characteristic order culminating in the upregulation of genes that determine terminal myeloid phenotypes at 80h. We also document two sharp transitions during differentiation, at 8h and 80h, when the global transcriptomic state changes quite suddenly. The first transition is shown to be associated with the initiation of metabolic remodeling and the second with the upregulation of myeloid phenotypic genes. Our analysis shows that information is transferred in successive cascading waves of gene expression during differentiation, punctuated by sharp transitions, each wave remodeling different aspects of the cells' physiology in a characteristic order.

## Results

### Differentiation reprograms PUER cells toward states resembling *in vivo* bone-marrow myeloid cells

PUER cells were isolated nearly three decades ago from the bone marrow of PU.1$^{-/-}$ mice and resemble macrophages and neutrophils morphologically, and in the expression of select markers, when differentiated [3,7,14]. The bulk RNA-Seq

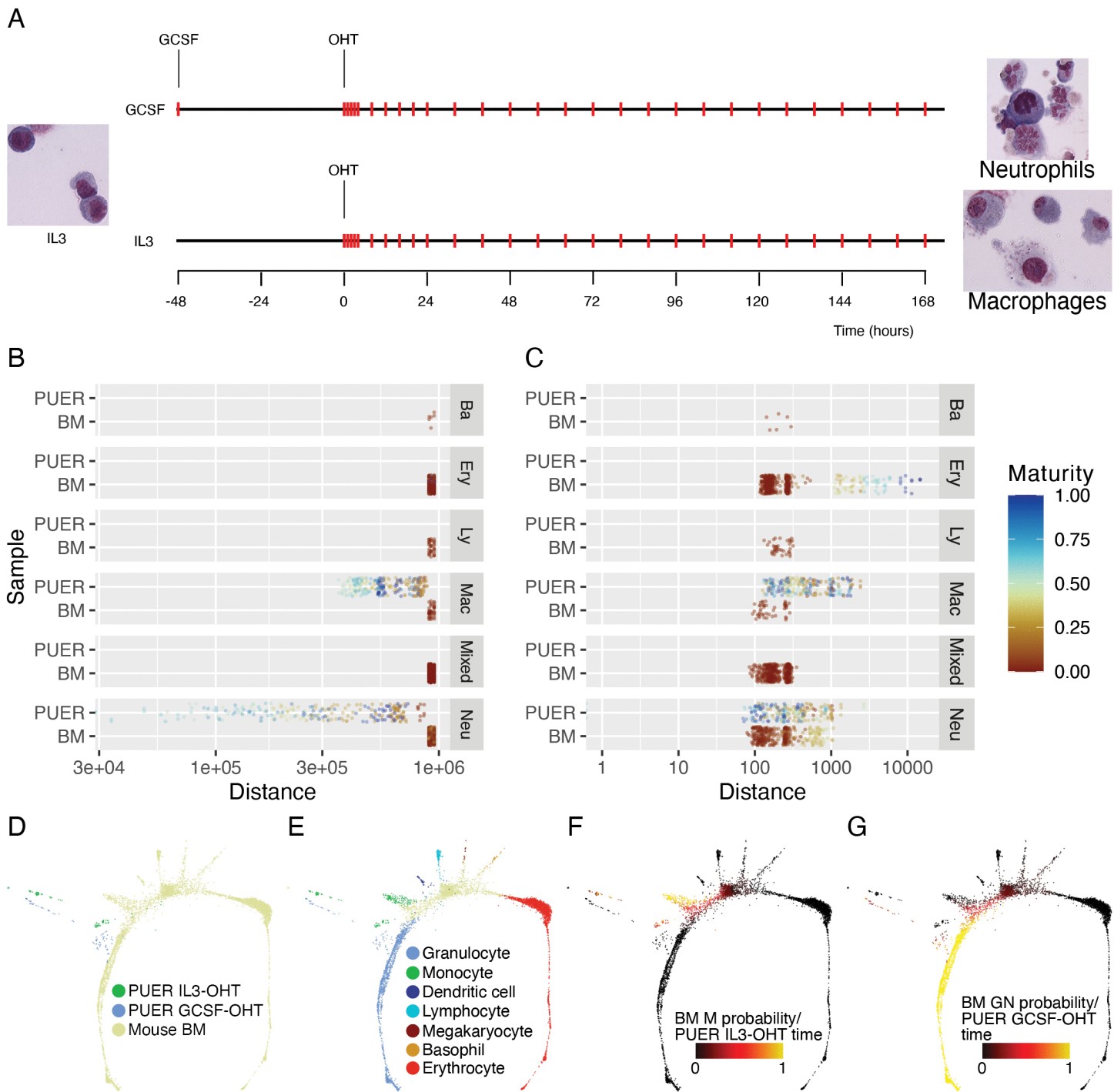

**Fig 1. *in vitro* PUER cell differentiation reprograms PU.1−/− cells to states resembling *in vivo* bone marrow macrophage and neutrophil progenitors. A**. Schematic showing the *in vitro* differentiation of PUER cells into macrophages and neutrophils. Red ticks indicate the timepoints at which cells were sampled for RNA-Seq. **B,C** The distribution of Euclidean distances between 88h GCSF-OHT samples and other PUER samples or *in vivo* BM single cells [16] either before (**B**) or after (**C**) the removal of batch effects. PUER cells in GCSF and IL3 conditions are labeled as Neu or Mac respectively. BM cells having a population balance analysis (PBA) [18] probability of more than 0.8 for a particular lineage are labeled as that lineage. Ba: basophil, Ery: erythrocyte, Ly: lymphocyte, Mac: monocyte, Neu: granulocyte. BM cells that have PBA probability less than 0.8 for all lineages are labeled as Mixed. The color of the points indicates developmental maturity so that more differentiated cells have higher values. For PUER cells, the developmental

maturity is given by the differentiation timepoint scaled to [0,1]. For BM cells, the developmental maturity is given by the additive inverse of the PBA potential scaled to [0,1]. **D–G** SPRING [19] plots of batch-corrected PUER samples and BM cells. **D**. Color shows PUER samples in IL3 (macrophage) or GCSF (neutrophil) conditions and BM cells. **E**. Color annotates PUER samples and BM cells by lineage. Only BM cells having a PBA probability of more than 0.8 for a particular lineage are annotated. **F**. BM cells are annotated by a heatmap of PBA probability for the monocyte (M) lineage. IL3-OHT PUER cells are annotated by a heatmap of differentiation timepoint. **G**. BM cells are annotated by a heatmap of PBA probability for the granulocyte (GN) lineage. GCSF-OHT PUER cells are annotated by a heatmap of differentiation timepoint.

data reported here, along with publicly available single-cell RNA-Seq (scRNA-Seq) datasets, presented an opportunity to quantitatively compare the transcriptomic state of differentiating PUER cells to those developing *in vivo*.

We chose an scRNA-Seq dataset of about 5,000 Kit+ mouse bone-marrow (BM) progenitors [16] to compare with the PUER data. The two datasets were acquired completely independently using starkly different experimental methodologies and a direct comparison is not possible. We registered the two datasets to each other by removing batch effects using MNNCorrect [17] treating each PUER sample as if it were a single cell (see Methods). Euclidean distance after cosine normalization of the genome-wide gene expression vectors [17] was used as a measure of the dissimilarity of transcriptomic state between samples or cells. Prior to batch correction, PUER samples are equidistant from all the different BM cell types even though the distances gradually increase with time within the PUER dataset (Fig 1B). This suggests that PUER samples are very distant from BM cells, potentially due to batch effects. After batch correction, the distances between PUER samples and BM cells vary over the same range as the distances between different PUER samples (Fig 1C), so that many PUER samples are closer to BM cells than they are to other PUER samples.

After batch correction, BM cells are among the 10 nearest neighbors of many, but not all, differentiated PUER samples (Fig 1C). Tusi *et al.* [16] used cell-fate specific gene expression signatures to assign each cell a probability to developing a particular cell fate. 88h after OHT induction in GCSF conditions (neutrophil differentiation) PUER samples are closest to BM cells with neutrophil probability greater than 0.8. Reflecting the hierarchical cell-fate relationships during hematopoiesis, the next closest cells are BM macrophages, followed by BM basophils, BM lymphocytes, and the most distant are BM erythrocytes.

We utilized SPRING plots [20], force-directed layouts of k-nearest neighbor (KNN) graphs of cells or samples, to visualize the overall relationship of differentiating PUER cells to BM cells. In the SPRING layout, differentiating PUER samples are placed at varying distances from the BM cells (Fig 1D). If the BM cells having more than 80% probability of achieving a particular fate are annotated, the PUER samples are closest to the macrophage/neutrophil junction, with GCSF-OHT PUER cells closest to BM neutrophils and IL3-OHT PUER cells closest to BM macrophages (Fig 1E). When the plot is annotated with measures of cell differentiation, time for PUER samples and fate probability for BM neutrophils (Fig 1F) or BM macrophages (Fig 1G), the undifferentiated and early time point PUER samples are the most distant from BM cells. PUER cells progressively approach BM cells at later time points. Furthermore, PUER cells approach BM cells having a neutrophil probability close to 1 during neutrophil differentiation and BM cells having macrophage probability >0.6 during macrophage differentiation.

The changes in global transcriptomic state are reflected in the expression patterns of the major known cell-fate markers and TFs [3,4,15,16,21]. The stem/progenitor genes *Kit*, *Flt3*, *Gata2*, *Sox4*, *Meis1*, *Mpl*, and *Mecom* are either downregulated or expressed at very low levels during PUER differentiation (S6 and S7 Figs). The sole exception is *Ly6a*, which is transiently upregulated before being downregulated. Markers that are expressed transiently along the neutrophil lineage in BM cells, *Ltf*, *Mpo*, *Elane*, *Gstm1*, *Prtn3*, and *Ctsg*, do the same during PUER neutrophil differentiation, while those expressed in the most mature BM neutrophils, *Lcn2*, *S100a8/9*, *Lyz2*, and *Fcgr3*, achieve their highest expression at the end of PUER differentiation (S8 Fig). Key neutrophil TFs, which are expressed transiently along the neutrophil lineage in BM cells are upregulated transiently in response to GCSF treatment of PUER cells and are expressed at higher levels than in macrophage conditions (S9 Fig). Among macrophage/monocyte markers, *Ly6c2*, *Ccr2*, *F13a1*, *Slpi*, *Adgre1*

(F4/80), and *Csf1r*, are all expressed during macrophage differentiation, although the last is expressed at relatively low levels. In IL3 conditions, these markers are either upregulated immediately upon OHT treatment or transiently downregulated before being upregulated. Interestingly, although these genes are regarded as monocyte markers, they exhibit similar expression patterns in GCSF conditions in PUER cells and are also expressed in neutrophil lineage cells in BM. While the macrophage markers *Itgam* (Cd11b) and *Mmp12* are upregulated in response to OHT treatment of PUER cells, they are not expressed in the monocyte lineage cells in BM, perhaps reflecting the relatively immature state of Kit + BM cells (S10 Fig). Finally, among the TFs implicated in macrophage differentiation, *Egr1*, *Egr2*, *Nab2*, *Zeb2*, and *Irf8* [3,4,22] are expressed in PUER cells and are upregulated relative to GCSF conditions in response to OHT treatment. However, *Irf8* is expressed at low levels and *Klf4* is undetectable, suggesting that macrophage differentiation in PUER cells occurs independently of the *Irf8*/ *Klf4* axis (S11 Fig).

These interrelationships between differentiating PUER cells *in vitro* and mouse BM cells developing *in vivo* support three conclusions. First, undifferentiated PUER samples are distant in their transcriptomic state from any BM cell, including progenitors. This can be understood to be a consequence of undifferentiated PUER cells being PU.1$^{-/-}$ mutants. Second, upon OHT induction, which rescues the PU.1$^{-/-}$ mutant, PUER cells approach BM cells in their transcriptomic state and in the expression of key markers and cell-fate genes. Third, differentiated PUER neutrophils and macrophages are closer in their transcriptomic state to neutrophils and macrophages developing *in vivo* respectively than to any other BM cell type. The differentiation of PUER cells can thus be seen as the reprogramming of cells from a mutant state to one very close to wildtype BM myeloid cells.

## PUER cells retain macrophage potential after GCSF pre-treatment

Given that reprogramming PUER cells into neutrophils requires the step-wise application of two treatments, GCSF and OHT, we wondered which of the two steps commits the cells to a neutrophil fate. We tested the fate potential of GCSF-pre-treated cells by switching them back to IL3 just prior to OHT induction and profiled their morphology (S12 Fig) and Cd117 (Kit), Cd11b, and Cx3cr1 expression (S13 Fig) after 96 hours. The distributions of cellular morphology and cell-surface marker expression of the switched cells were indistinguishable from those of 96h IL3-OHT cells. It is possible that the population retaining monocyte potential after GCSF pre-treatment and generating monocytes upon switching to IL3 is a small proportion of the total, while the majority of the cells are irreversibility committed to the neutrophil fate. In such a scenario, one would expect the latter population to die, resulting in low viability. The viability of the switched cells at 96h was 93%, similar to that of 96h IL3-OHT cells, suggesting that this is not the case. While conclusively establishing whether individual cells remain bipotential after GCSF pre-treatment would require single-cell lineage tracing, the switch experiment supports the view that the effects of GCSF pre-treatment are reversible and that it is PU.1 induction by OHT treatment that commits PUER cells to their fate.

## Global changes in the gene expression landscape during myeloid differentiation

As a first step we sought to uncover temporal patterns of changes in genome-wide gene expression. We computed Pearson correlation of genome-wide gene expression between all pairs of time points (Fig 2A and 2B). As one would expect, the correlation coefficient is higher for nearby time points and reduces as the difference between the time points increases. The largest effect is that of GCSF pre-treatment (−48h vs. 0h; Fig 2A), which can be understood as the result of the large time difference of 48 hours and the important role that GCSF plays in the growth and maturation of hematopoietic cells [23].

Unexpectedly, we found that time points could be divided into at least two groups, early (<12 hours) and late (≥12 hours), so that timepoints have much higher correlation within each group than to timepoints from the other group. This is most easily discerned in the macrophage differentiation in IL3 conditions (Fig 2B). For example, the 16h timepoint has very low correlation, $0 \geq r < 0.2$, with the 2h timepoint in the early group, occurring only 14 hours earlier, but has very high

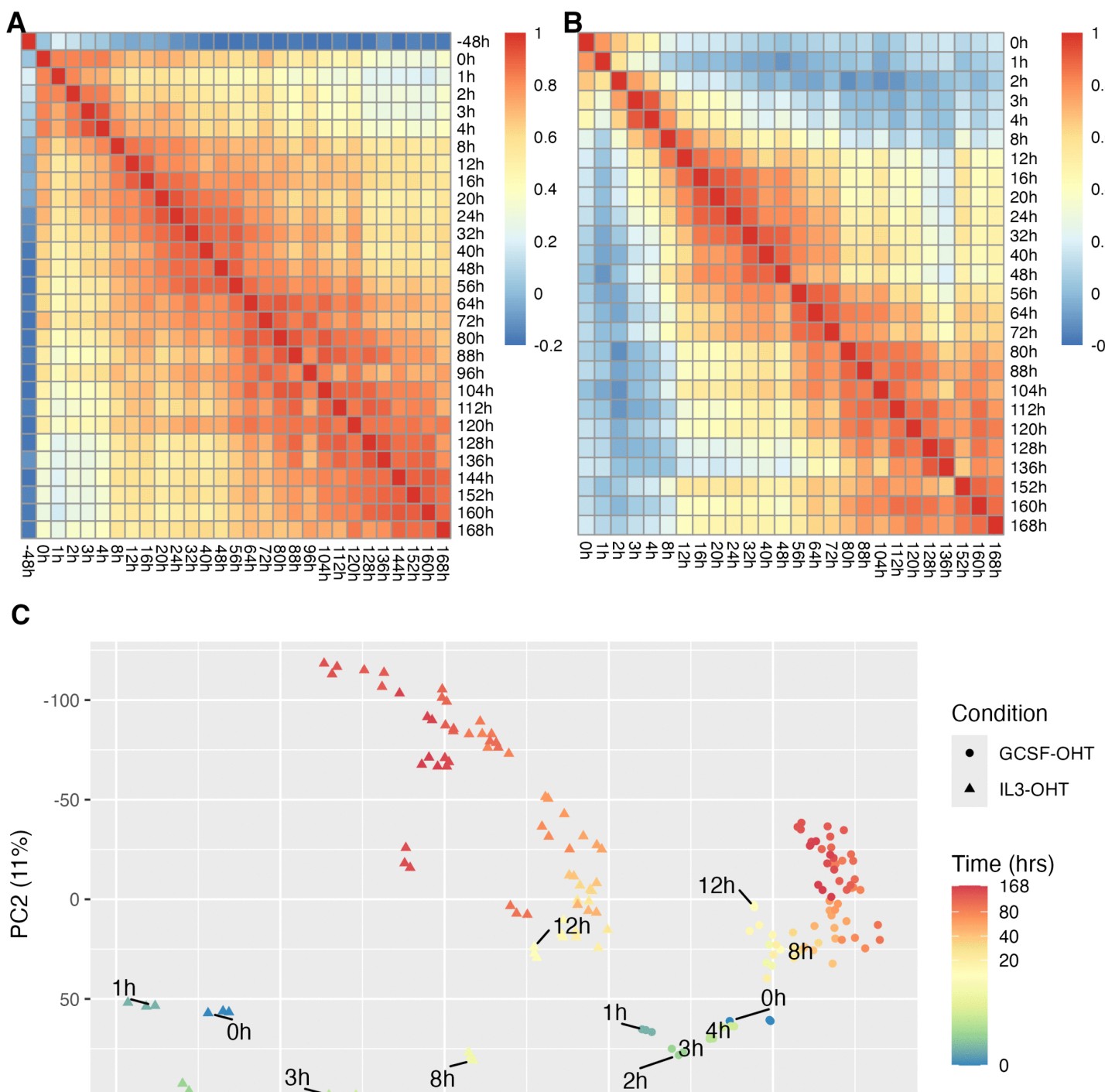

**Fig 2. PUER cell differentiation is punctuated by two sharp transitions. A,B**. Pearson correlation coefficient between genome-wide gene expression at each pair of time points in GCSF (**A**) or IL3 (**B**) conditions during PUER differentiation (Fig 1A) is plotted as a heatmap. **C**. Principal components analysis of all the samples. Gene expression was standardized to have zero mean and unit variance. The samples are plotted along the first two principal components, PC1 and PC2, that account for 25% of the total variance.

correlation, $r > 0.8$, with the similarly spaced 32h timepoint in the late group. Although post-OHT timepoints have higher correlation overall, a similar pattern is discernible in the GCSF condition. That the differentiation can be divided into two phases suggests that there is a large-scale transition in genome-wide gene expression patterns occurring around $8 - 12$ hours.

We further characterized the global patterns of gene expression time evolution using principal component analysis (PCA) (Fig 2C). The first two principal component axes accounted for 25% of the total variance in the data, hinting that there are two main effects and that the two-dimensional space is a good approximation to the high-dimensional gene expression space.

The GCSF and IL3 conditions follow distinct differentiation trajectories and are separated by a large shift along the PC1 axis, occurring during the GCSF pre-treatment, which implies that the first principal component corresponds to the effect of GCSF. In both the GCSF and IL3 conditions, the trajectories move along the PC2 axis after OHT addition at 0h, suggesting that the second principal component corresponds to the effect of PU.1 and time. The displacement between the 8h and 12h IL3 time points is the second largest after that of GCSF pre-treatment, which implies a very large rate of change in genome-wide gene expression given that it occurs in only 4 hours compared to the 48 hour duration of GCSF pre-treatment. Similar but smaller jumps are observed between the 4h and 8h GCSF and the 72h and 80h IL3 time points, while there is a sharp reversal of the direction of movement at the 136h IL3 timepoint. These jumps corroborate the inference drawn from the Pearson correlation analysis (Fig 2A,B) that there is a large-scale transition in the pattern of genome-wide gene expression around $8 - 12$ hours after OHT induction, and indicate that there are other such transitions occurring at later stages of the differentiation as well.

## Diversity of temporal patterns of transient gene expression

In order to gain insight into the genome-wide transitions occurring during the course of differentiation we next analyzed the temporal patterns of the expression of individual genes. We enriched for genes likely to be regulated during the differentiation process by first identifying genes expressed differentially between the end points, undifferentiated PUER cells, −48h GCSF or 0h IL3, and 7-day OHT treated samples, 168h GCSF or IL3 (Fig 3A and 3D). 43% and 27% of genes were differentially expressed between the end points in GCSF and IL3 conditions respectively. The neutrophil differentiation is the compounded effect of GCSF and OHT treatments and one may discern between the two by identifying genes differentially expressed because of GCSF pre-treatment, by comparing −48h samples to 0h GCSF samples, and those differentially expressed before and after 7 days of OHT treatment, by comparing 0h to 168h GCSF samples (Fig 3B and 3C). Consistent with both the correlation analysis and PCA, more genes are differentially expressed due to GCSF pre-treatment (37%) than due to OHT treatment (29%), even though the latter is conducted over a larger time interval. There is significant overlap in the genes differentially expressed in the two conditions. Of the 2,029 genes downregulated between 0h and 168h in IL3 conditions 1,370 (68%) are also downregulated between −48h and 168h in GCSF conditions (Fig 3E). Similarly 61% of the genes upregulated between the IL3 endpoints are also upregulated between the GCSF endpoints (Fig 3F).

The temporal expression patterns of the differentially expressed genes are very diverse and show extensive transient regulation, in which expression at the start and end of differentiation is similar but is modulated in the middle (S14 Fig). In order to better reveal broader patterns, the genes were clustered hierarchically according to the similarity of their temporal expression patterns. Several patterns are noticeable. Consistent with all the previous analyses, GCSF pre-treatment exerts significant effect on the gene expression, with a large number of genes turning off and a smaller but still sizable group turning on at 0h in the GCSF treatment (S14A Fig). Another significant shift in the gene expression occurs around the $8 - 12$h timepoints, when a large number of genes are upregulated and a smaller number of genes are downregulated. Furthermore, the number of genes coordinately regulated in this manner is greater in the IL3 condition than the GCSF condition. Also discernible in the IL3 condition, but less so in the GCSF condition, are several waves of transient

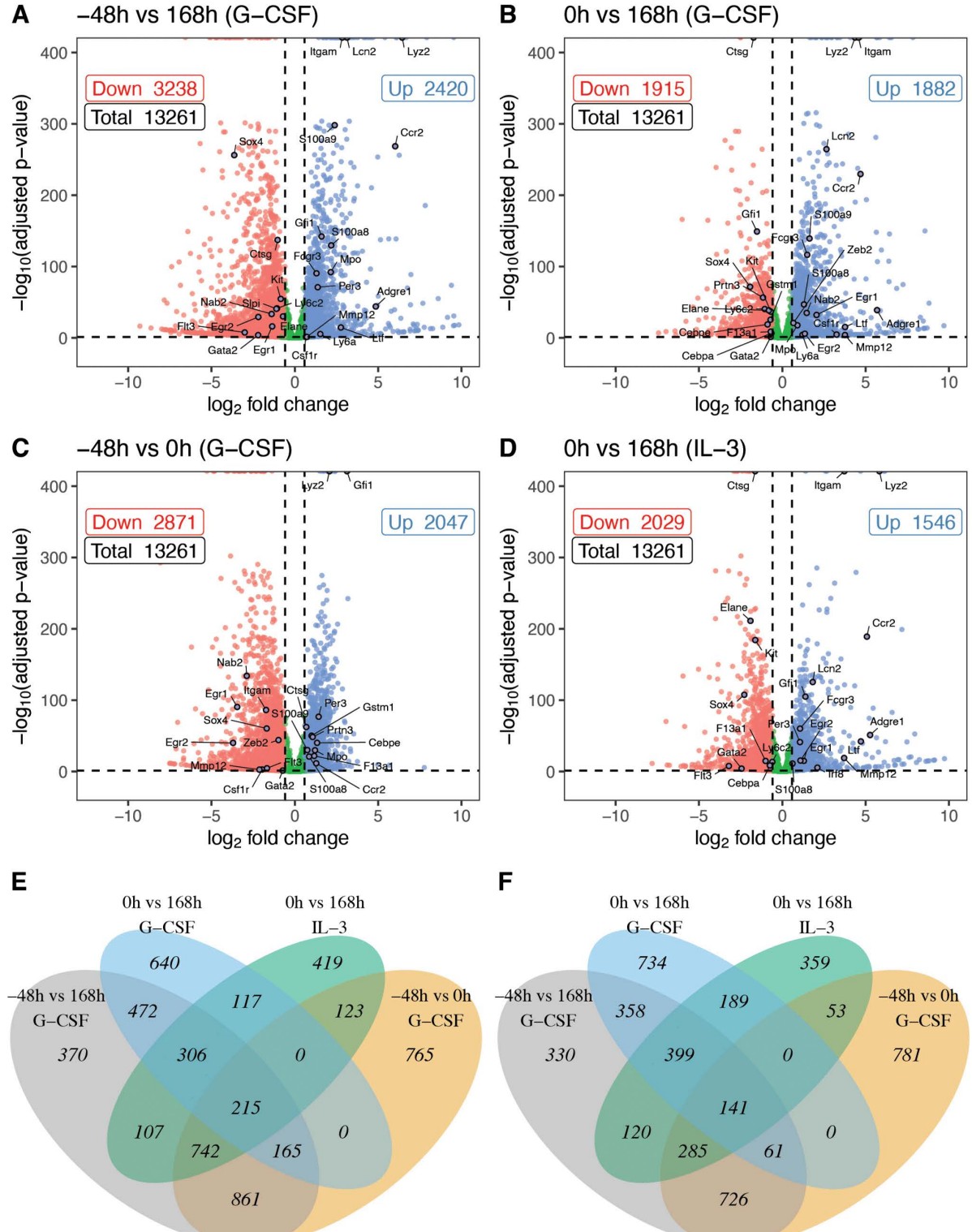

**Fig 3. The identification of genes expressed differentially between the endpoints of the differentiation.** Gene expressed differentially between the endpoints of PUER differentiation (Fig 1A). **A–D**. Scatter plots of *p*-value vs. fold change for all the genes. The *p*-value and fold change thresholds used to identify DEGs (see Methods) are shown as horizontal and vertical dashed lines respectively. Key differentially expressed lineage markers and TFs

have been annotated. **E–F**. Venn diagrams showing the intersection of the different sets of DEGs identified in this analysis. **A**. Comparison of undifferentiated PUER cells (−48h) with cells treated with OHT for 7 days in GCSF conditions (168h). **B**. Comparison of PUER cells pre-treated with GCSF for 48 hours (0h) with cells treated with OHT for 7 days in GCSF conditions (168h). **C**. Comparison of undifferentiated PUER cells (−48h) with those pre-treated with GCSF for 48 hours (0h). **D**. Comparison of undifferentiated PUER cells (−48h) with those treated with OHT for 7 days in IL3 conditions (168h). **E,F**. Overlap of DEGs for selected time points for down-regulated (E) and up-regulated (F) genes.

gene upregulation and downregulation during the first 8 hours of differentiation, with different groups of genes peaking at different timepoints. Finally, a transition at 80h in the IL3 condition when a smaller number of genes are either up- or down-regulated can also be seen. To summarize, the temporal gene expression patterns further corroborate the transitions inferred previously, show that the first $8 - 12$ hours of differentiation involve rapid changes in expression, and reveal extensive transient and coordinated regulation of groups of genes.

The above analysis ignores genes that are transiently expressed during differentiation but revert to their original expression by the end and are not identifiable as DEGs between the end points. Focusing only on DEGs between endpoints, therefore, could potentially ignore important aspects of the information transfer process of differentiation. We sought a method to identify and classify genes based on the entire temporal pattern of expression instead of the behavior at the end points. Non-negative matrix factorization (NMF) [24,25] is an unsupervised learning method that simultaneously performs dimensionality reduction and clustering. Given the expression of gene $g$ at time $t$, $X_{gt}$, we used NMF to find $M$ gene expression patterns or *behaviors*, $H_{mt}$, and weights, $W_{gm}$, so that the expression of the gene is approximated as a weighted sum of the $M$ behaviors, $X_{gt} \approx \sum_{m=1}^{M} W_{gm}H_{mt}$. The behaviors and weights were restricted to be non-negative so that a behavior is interpretable as the expression of a metagene. Dimensionality reduction was achieved by choosing the number of behaviors, $M$, to be much smaller than the total number of genes. One important capability of NMF is that it performs automatic feature identification; the behaviors are spatially or temporally restricted and correspond to parts of the overall pattern. NMF applied to photographs of faces, extracts components that correspond to interpretable facial features, such as noses and eyes [24]. NMF on temporal gene expression patterns therefore is expected to identify temporally restricted features such as pulses and oscillations.

We found that the error between successive NMF approximations has a local minimum at $M=7$ (see Methods; S4 Fig) and chose $M=10$, with a slightly higher error, to ensure that the full diversity of temporal expression patterns was accounted for. NMF was performed on the GCSF and IL3 time points together so that any constraints on gene expression across conditions, such as a gene that is upregulated by GCSF but downregulated by IL3, were reflected in the behaviors (Fig 4 and Table 1). Performing NMF on unscaled data yielded behaviors that were very similar to the ones inferred with maximum gene expression scaled to 1 (S15 Fig) and all subsequent analysis is based on scaled data. With the exception of three behaviors, 3, 5, and 8, all behaviors showed mutually exclusive expression patterns so that behaviors regulated in the GCSF condition were not regulated in IL3 and vice versa. We may infer, for example, that behavior 6 genes respond to OHT treatment in IL3 but are unaffected in GCSF. In behavior 5, expression is downregulated in both GCSF and IL3 conditions over the first two time points after the initiation of treatment, which occurs 48 hours after substituting GCSF for IL3 in neutrophil differentiation and in the first two hours following OHT addition in macrophage differentiation. Behavior 8 is a pulse of expression starting around 8h after OHT addition and terminating around 80h in both conditions. Behavior 3 exhibits a pulse of expression from 2h to 48h followed by oscillatory expression lasting to the end of the differentiation.

Three behaviors, 1, 2, and 4, have expression during neutrophil differentiation but not during macrophage differentiation. Behavior 1 expression is upregulated in response to GCSF pre-treatment but is then downregulated upon OHT induction. Behavior 2 is also upregulated by GCSF and transiently downregulated by OHT, but is re-upregulated once more, peaking at 100h. In behavior 4, expression initiates around 100h and remains on until the end of the differentiation. Three macrophage specific behaviors, 6, 7, and 9, have pulsatile expression lasting between $0 - 8h$, $2 - 48h$, and $12 - 128h$ respectively. Behavior 10 has expression that initiates around 80h and remains on until the end of the differentiation. The process

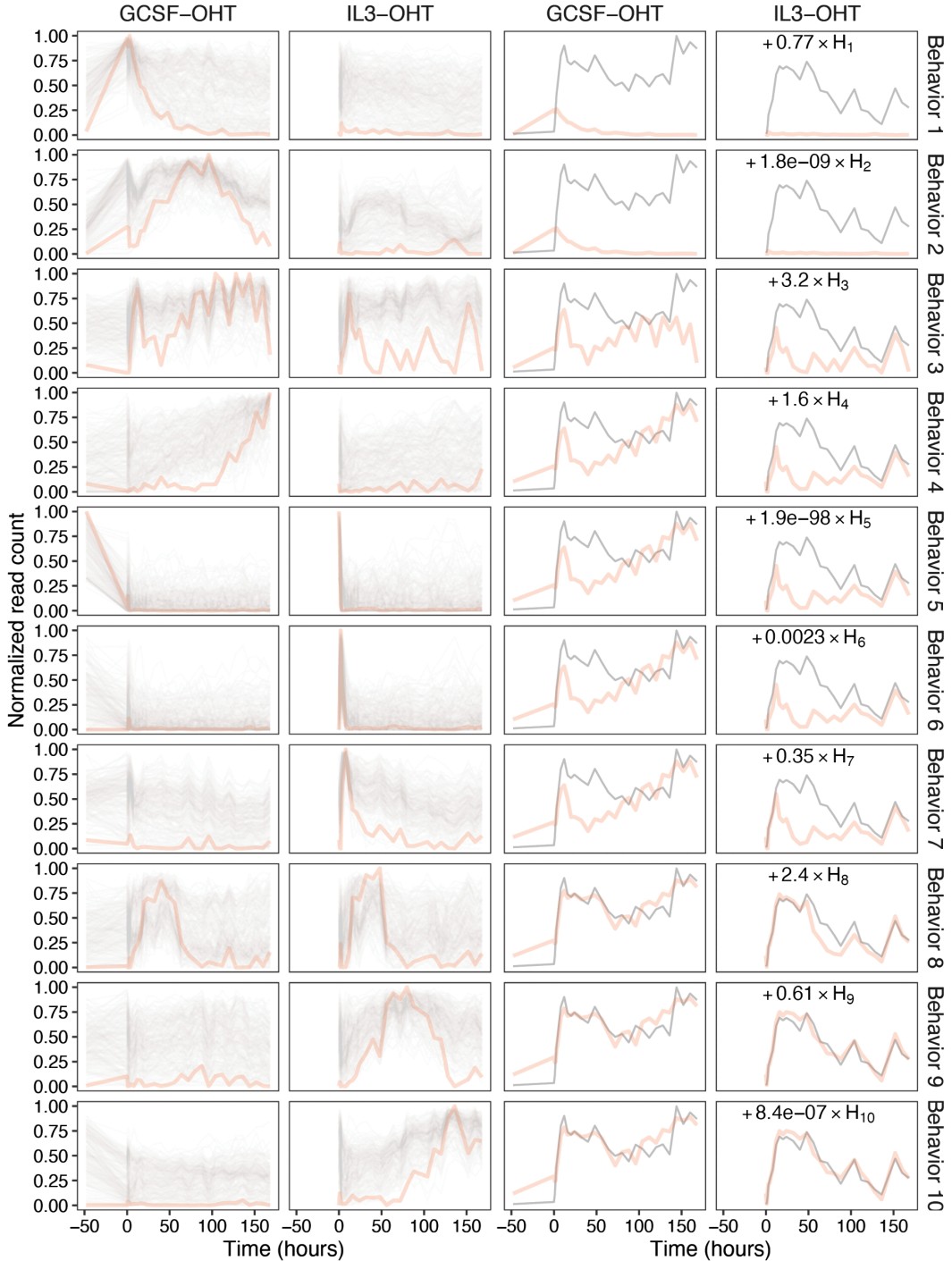

**Fig 4. The temporal behaviors of the 10 metagenes identified by NMF.** The behaviors inferred from the genome-wide gene expression time series data (Fig 1A) using NMF are shown in the first two columns. The temporal behavior of each metagene is shown in red. The temporal expression patterns, with maximum expression scaled to 1, of the transcripts having the highest 200 weights for each behavior are shown in black. The last two columns illustrate the reconstruction of the expression pattern of *Cx3cr1* by the sequential accumulation of the weighted behaviors $\sum_{1}^{M} W_{gm}H_{m}(t)$, where $M$ is the row number, $H_{m}(t)$ is the behavior of metagene $m$, and $W_{gm}$ is *Cx3cr1*'s weight for the $m$th behavior. *Cx3cr1*'s expression pattern is shown in black and the weighted sum accumulating sequentially from the top to bottom is shown in red.

PLOS Computational Biology

**Table 1. Summary of biological processes enriched in the top 500 genes for each behavior. The top 30 enriched terms (S26 Fig–S35 Fig) are summarized. Terms in italics were not in the top 30.**

| Meta-gene | Behavior in GCSF | Behavior in IL3 | Enriched biological processes |
|---|---|---|---|
| 1 | Upregulated by GCSF and downregulated by OHT | Not regulated | Cell cycle, metabolism |
| 2 | Upregulated by GCSF, transiently downregulated by OHT, peaking around 100h | Not regulated | Cell cycle |
| 3 | Pulse from 2h to 48h, re-upregulated and expressed in an oscillating pattern until the end | Pulse from 2h to 48h, oscillations at later timepoints | Autophagy, metabolism, cytoskeleton, NF-kB signaling |
| 4 | Upregulated after 100h and peaks at the end | Not regulated | Granulocyte/neutrophil migration/chemotaxis, innate immune response |
| 5 | Downregulated by GCSF pre-treatment | Downregulated by OHT by 2h | Extracellular signaling, epithelial-to-mesenchymal transition, thermogenesis, cell adhesion, cytoskeleton, migration |
| 6 | Not regulated | Pulse from 0h to 8h | TGFb, Erk1/2, Ras signaling, cytoskeleton, extracellular matrix |
| 7 | Not regulated | Pulse from 2h to 48h | Translation, mRNA processing |
| 8 | Pulse from 8h to 80h | Pulse from 8h to 80h | Metabolism, leukocyte activation |
| 9 | Not regulated | Pulse from 12h to 128h | Metabolism |
| 10 | Not regulated | Upregulated after 80h and peaks at the end | Translation, metabolism, *myeloid/leukocyte proliferation*, *innate immunity* |

of information transfer during differentiation thus appears to involve multiple successive waves of gene expression with handover occurring at time points that correspond approximately to the global transitions at 8 – 12h (behaviors 6, 8, 9) and 80 – 100h (behaviors 4, 9, 10) and culminating in the switching on of a subset of genes permanently.

It is possible that these behaviors do not correspond to parts of the expression of any one gene but are used by NMF to construct entirely different patterns by linear combination. We checked whether this was true by visualizing genome-wide gene expression after sorting transcripts according to their dominant—highest weighted—behavior (Fig 5). About 8,000 transcripts are not expressed at all and have zero weight for all the behaviors. Another group of transcripts are expressed but their expression does not vary over time. These transcripts—particularly enriched among behavior 2 and 3 clusters—have high weights for multiple behaviors. Excepting the unexpressed and constantly expressed groups, the remaining transcripts are dominated by a few behaviors so that their expression patterns match one or more behaviors. Furthermore, the pairwise dissimilarity (distance) matrix has significant block structure (S16 Fig), showing that the dominant behavior alone can account for most of the observed similarity in temporal expression patterns between genes. Clustering genes hierarchically (S17 Fig) yields about 6 clusters with similar expression patterns as the behaviors, although they are not so distinct as when clustering on dominant behavior. This could be because genes expressed early in a pulsatile manner, having relatively low correlation with all other genes, might get grouped in an unpredictable manner. The behaviors inferred by NMF therefore are exhibited as real observable gene expression patterns and the method is better at grouping and revealing transient expression than hierarchical clustering.

We also checked whether NMF is applicable more generally to other gene expression datasets. We inferred behaviors in Tusi *et al.*'s Kit+ mouse BM progenitor scRNA-Seq dataset [16]. Successively increasing the number of metagenes, we found that no new behaviors were revealed when performing NMF with more than 17 metagenes. The algorithm inferred behaviors corresponding to the erythroid (behavior 14), neutrophil (2), monocyte (17), lymphoid (11), and basophil (5) lineages (S18 Fig). Additional behaviors corresponding to intermediate states such as early erythrocyte (behavior 9), early neutrophil (13), granulocyte-monocyte (12), megakaryocyte-erythrocyte (15) are also discernible. Notably, these behaviors match Tusi *et al.*'s marker-based lineage assignments even though NMF inferred them without any foreknowledge of

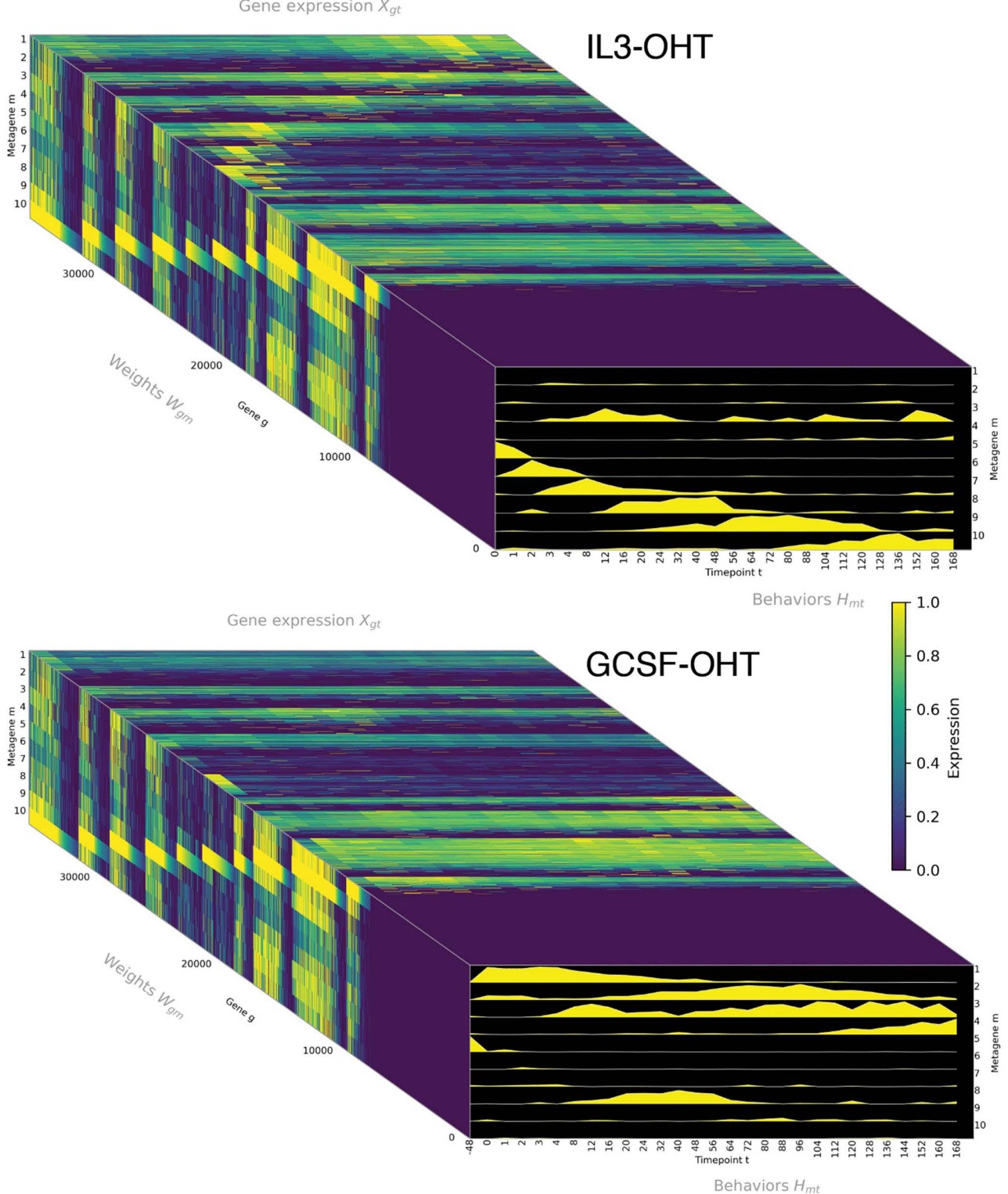

**Fig 5. Clustering of genome-wide gene expression by temporal behaviors.** The gene expression matrix ($X_{gt}$) from the genome-wide time series data (Fig 1A), the temporal behaviors of the 10 metagenes ($H_{mt}$)), and the weights of each transcript for each behavior ($W_{gm}$) as determined by NMF are

depicted on the faces of a rectangular prism. $X_{gt}$ and $W_{gm}$ are heatmaps with color representing scaled gene expression or weights respectively. The behaviors are plotted as stacked lineplots with time on the *x*-axis and expression on the *y*-axis. Transcripts having the same dominant behavior were grouped together and then sorted in descending order of the weight of the dominant behavior. IL3-OHT and GCSF-OHT samples are shown on the top and bottom prisms respectively.

lineages or developmental stages. Highlighting the importance of scaling the data, NMF on unscaled single-cell data (S19 Fig) infers behaviors corresponding to only three lineages: late erythrocyte, late neutrophil, and progenitors. NMF infers fewer distinct behaviors from unscaled data most likely because, given the long-tailed distribution of genome-wide gene expression, a few genes with very high expression contribute disproportionately to the error of the approximation and dominate the factorization process.

### Gene expression changes most rapidly during the earliest stages of the differentiation

Temporal gene expression patterns of transcripts differentially expressed between the endpoints and the pulsatile behaviors found by NMF suggested that most of the changes in genome-wide gene expression are concentrated in the first 12 hours of the differentiation. We checked whether this is true by quantifying the pace of gene expression changes during differentiation. We determined the genes differentially expressed between consecutive timepoints and utilized the number of such genes as a measure of the rate of change (Fig 6A). The timing of sharp gene expression shifts during the differentiation may indicate important moments when crucial lineage decisions are made. Most DEGs between consecutive time points are detected before the 12h timepoint in both conditions. This confirms that rapid, early changes during the first 12 hours after OHT induction are a general feature and not just restricted to genes differentially expressed between the endpoints of the differentiation. Additionally, there is very little overlap between DEGs detected between different pairs of consecutive timepoints, implying that genes are undergoing rapid but short-lived, transient changes during the early stages of the differentiation (Fig 6B and 6C).

We next considered to what extent rapid early changes could be attributed to direct regulation by PU.1. We utilized CUT&RUN to profile PU.1 binding genome-wide in the undifferentiated and 48h IL3 samples. 6,724 statistically significant (FDR ≤ 0.05) peaks were detected in the differentiated sample relative to the undifferentiated one, which is expected to have none or very little PUER activity. We determined the enrichment of PU.1 binding within either ±1kb or ±50kb of the transcription start site (TSS) of the genes differentially expressed between consecutive time points in IL3 conditions (S20 Fig). With the exception of PU.1 binding in proximal regions of DEGs between 1h and 2h and 3h and 4h, the DEGs between consecutive early time points up to 12h are significantly enriched in PU.1 binding ($p \leq 2.6 \times 10^{-2}$; one-sided Fisher's exact test) both near the TSS and in distal regions. The fraction bound to proximal regions of the DEGs is ~1.34-fold to ~1.82-fold higher compared to background while the fraction bound to distal regions of DEGs is ~1.14-fold to ~2.26-fold higher. The enrichment in PU.1 binding declines when comparing later consecutive timepoints, approaching ~1 for the 160h-168h comparison, supporting the interpretation that rapid early changes are likely a consequence of direct PU.1 regulation. More upregulated genes are bound by PU.1 and have higher enrichment than downregulated ones, consistent with PU.1's role as a transcriptional activator.

We also checked whether the genes exhibiting particular behaviors were enriched for PU.1 binding or not. We identified the genes having the highest 500 weights for each behavior and determined whether PU.1 was bound within either ±1kb or ±50kb of their TSSs (S21 Fig). The top 500 genes for behaviors 3 and 4 are enriched in PU.1 binding within ±50kb of the TSS ($p \leq 2.01 \times 10^{-5}$), while behaviors 2, 3, 4, and 10 are enriched in PU.1 binding within ±1kb ($p \leq 2.99 \times 10^{-2}$). We limit the analysis to the subset of these behaviors having expression in IL3 conditions, behaviors 3 and 10, since it is not possible to infer a regulatory relationship between behaviors expressed exclusively in GCSF conditions, behaviors 2 and 4, and PU.1 based on binding data from the IL3 condition. Behavior 3 features an early pulsatile response in IL3, while behavior 10 genes are turned on at late timepoints in IL3. This implies that PU.1 tends to regulate genes upregulated transiently at early timepoints or those turned on permanently at the end. The apparent contradiction with the lack of enrichment in

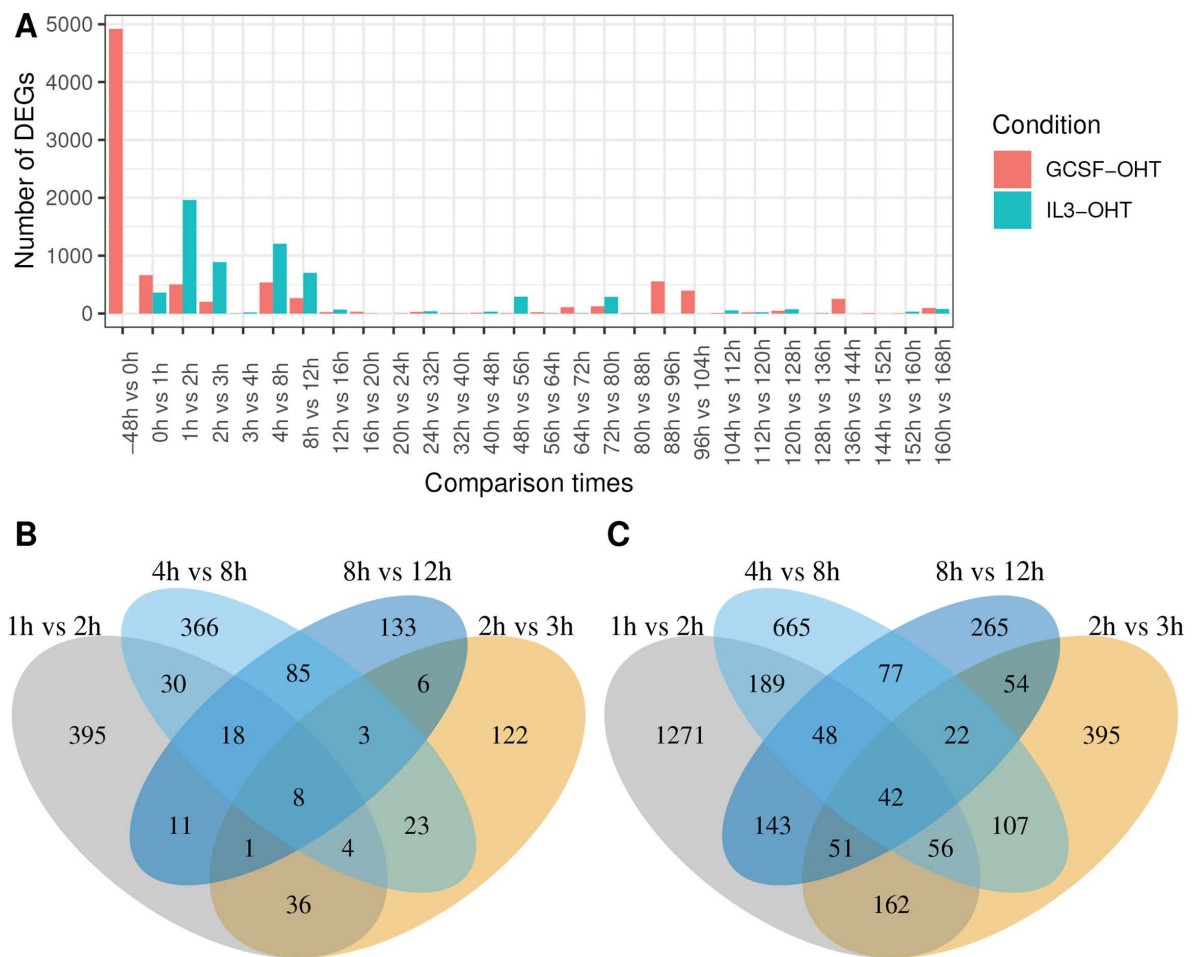

**Fig 6. The number of DEGs between consecutive time points. (A)** The number of DEGs detected between each pair of consecutive time points (Fig 1A). The overlap between the DEGs detected at different time points in the **(B)** GCSF or the **(C)** IL3 conditions.

late DEGs could be because the genes upregulated at the end have a slower rate of change and may not be changing in a statistically significant manner between consecutive late timepoints. Behaviors 1, 5, 6, 7, and 8 are not enriched for PU.1 binding whether proximally or distally. Given the nature of the behaviors (Table 1), this implies that PU.1 tends not to regulate GCSF responsive genes (behavior 1), genes downregulated by GCSF or OHT (behavior 5), or genes transiently upregulated in the middle of the differentiation (behaviors 7 or 8).

## Functional enrichment analysis

We performed functional enrichment analysis to determine the biological processes induced during differentiation. In one approach, we performed this analysis for DEGs detected between consecutive early time points. In a complementary approach, we determined gene ontology (GO) terms enriched for the top 500 genes for each of the ten behaviors.

## DEGs between consecutive early time points

The GO terms enriched in the DEGs ascertained between consecutive early time points (Fig 7, Fig 8, S22 Fig–S25 Fig) show clear temporal patterns. The top 30 GO terms—having the lowest 30 adjusted *p*-values—enriched in the first comparison, 0h

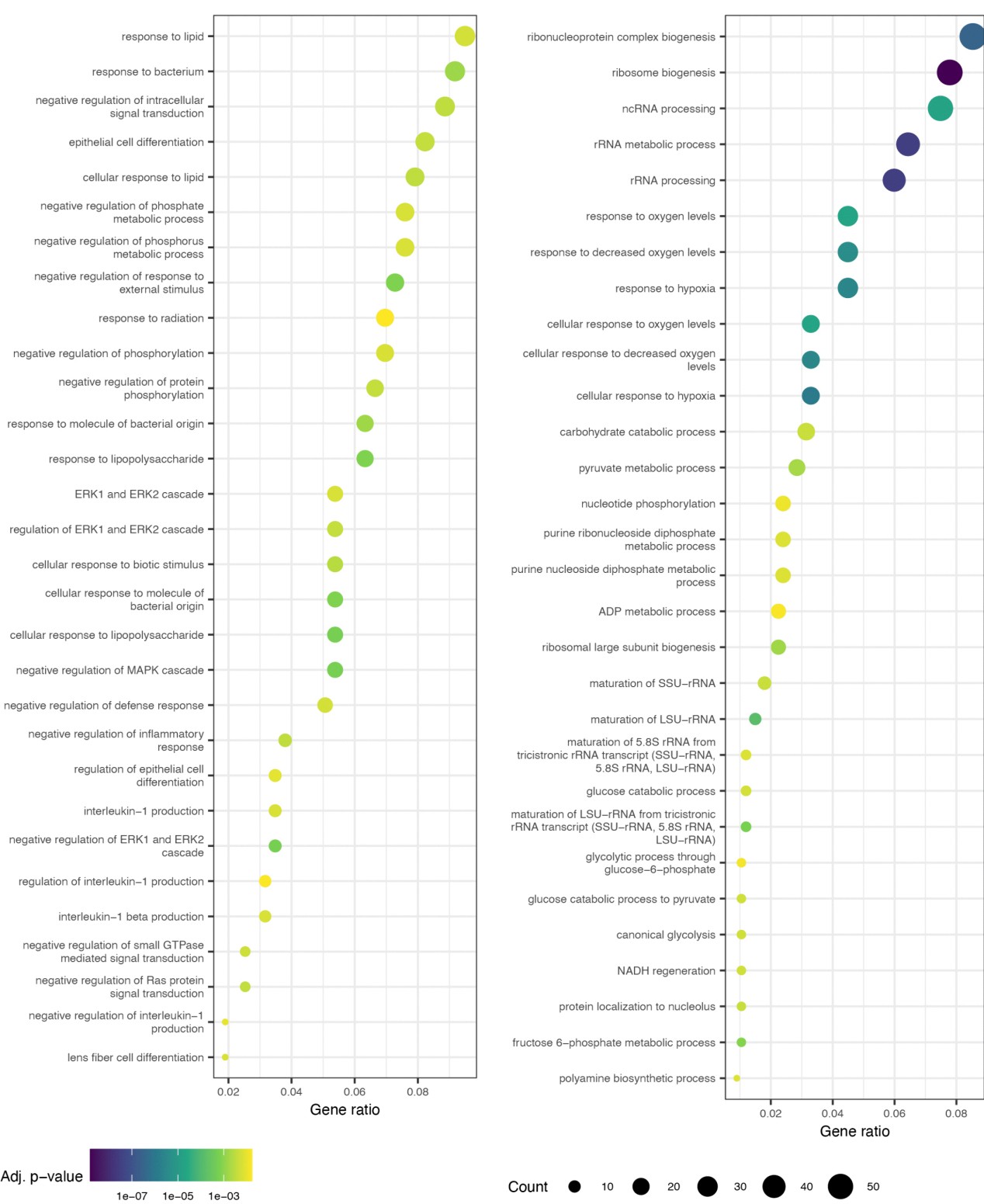

**Fig 7. GO terms enriched for genes differentially expressed between 0h and 1h or 8h and 12h IL3 samples.** GO terms enriched for DEGs between 0h and 1h and between 8h and 12h are shown on the left and right respectively. The Benjamini-Hochberg adjusted *p*-value for each enriched GO term is indicated by the color of the point. The number of DEGs overlapping with a term are depicted with the size of the point. The *x*-axis is the fraction of the DEGs overlapping with a GO term (gene ratio).

vs. 1h (Fig 7 and 8), are dominated by terms for intracellular signaling relating to MAPK, Erk1/2, and other pathways in both IL3 and GCSF conditions. This suggests that PU.1 activation by OHT very rapidly induces signaling cascades. In the 1h vs. 2h comparison, there are fewer signaling-related terms while terms related to development and cellular differentiation, especially leukocyte differentiation, are prominent. This suggests that the cell-fate decision is made very early during the differentiation process. Although biological functions related to myeloid differentiation are common, GO terms related to the differentiation of non-myeloid cell lineages are also enriched, reflecting the pleiotropic roles that most hematopoietic TFs play in multiple lineages.

The GO terms related to signal transduction, development, and cellular differentiation are no longer present in the top 30 for comparisons between 4h and 8h (S24 and S25 Figs) or between 8 and 12h (Fig 7 and 8). Instead, GO terms related to physiological processes such metabolism, ribosome biogenesis, ion homeostasis, and cell-cell adhesion were highly ranked. This suggests that the cells initiate the remodeling of physiological processes as early as 8h in the course of the differentiation to implement a decision that's already been made. Furthermore, this analysis identifies the $8 - 12h$ transition observed above (Fig 2 and S14 Fig) as the initiation of large-scale physiological changes in response to differentiation cues. Finally, comparisons between later timepoints were also not enriched in myeloid differentiation and cell-fate commitment GO terms but were instead linked to various physiological processes. These results, paired with the finding that GCSF pre-treatment does not cause lineage commitment (S12 and S13 Figs), strengthens the interpretation that the cell-fate decision is made within the first 4 hours of OHT induction.

### Top 500 genes for each behavior

The comparison of consecutive time points is limited to the detection of fast changing biological processes since genes changing expression at a slower pace may only build up statistically significant differences over larger time spans. In order to understand the progression of biological phenomena over the entire span of the differentiation, we performed GO analysis on the genes having highest 500 weights for each behavior (S26 Fig–S35 Fig, Table 1).

Mirroring the inference of the DEG GO analysis, the earliest acting behavior in IL3, behavior 6, expressed in a pulsatile manner during the first few hours, is enriched in signal transduction GO terms. The next wave of gene expression lasts from 2h to 48h (behavior 7) and involves translation and mRNA processing. This is followed by a third wave lasting from about 8h to 128h (behaviors 8 and 9) that is enriched in GO terms for metabolism and leukocyte activation. The process culminates in a set of genes that are upregulated around 80h and peak in expression at the end of the differentiation. This final IL3 wave is enriched in GO terms for myeloid proliferation and innate immunity. The GCSF condition behaviors follow a pattern similar to that of IL3. Behavior 3, which features a wave of expression from 2h to 48h followed by re-upregulation and expression until the end of the differentiation, is enriched in extracellular signaling GO terms, similar to the early pulse behavior 6, and also metabolism terms, similar to late behaviors 9 and 10 in IL3. Behavior 8, a wave of expression from 8h to 80h, is enriched in metabolism and leukocyte activation terms. A final wave of gene expression starting at about 100h and peaking at the end of the differentiation is enriched in granulocyte phenotypes such as migration, chemotaxis, and immune response.

Besides these successive waves of gene expression induced by OHT treatment, behavior 5 features genes that are downregulated in response to both GCSF and OHT treatment. This behavior is enriched in an eclectic set of terms, extracellular signaling, epithelial-to-mesenchymal transition, thermogenesis, cell adhesion, cytoskeleton, and migration. This might reflect the suppression of non-myeloid cell fates by GCSF and PU.1. Behavior 1, which is specific to GCSF, features genes that are transiently upregulated in response to GCSF and downregulated by OHT induction is highly enriched for cell cycle GO terms, implying that GCSF and OHT have opposing effects on the cell cycle.

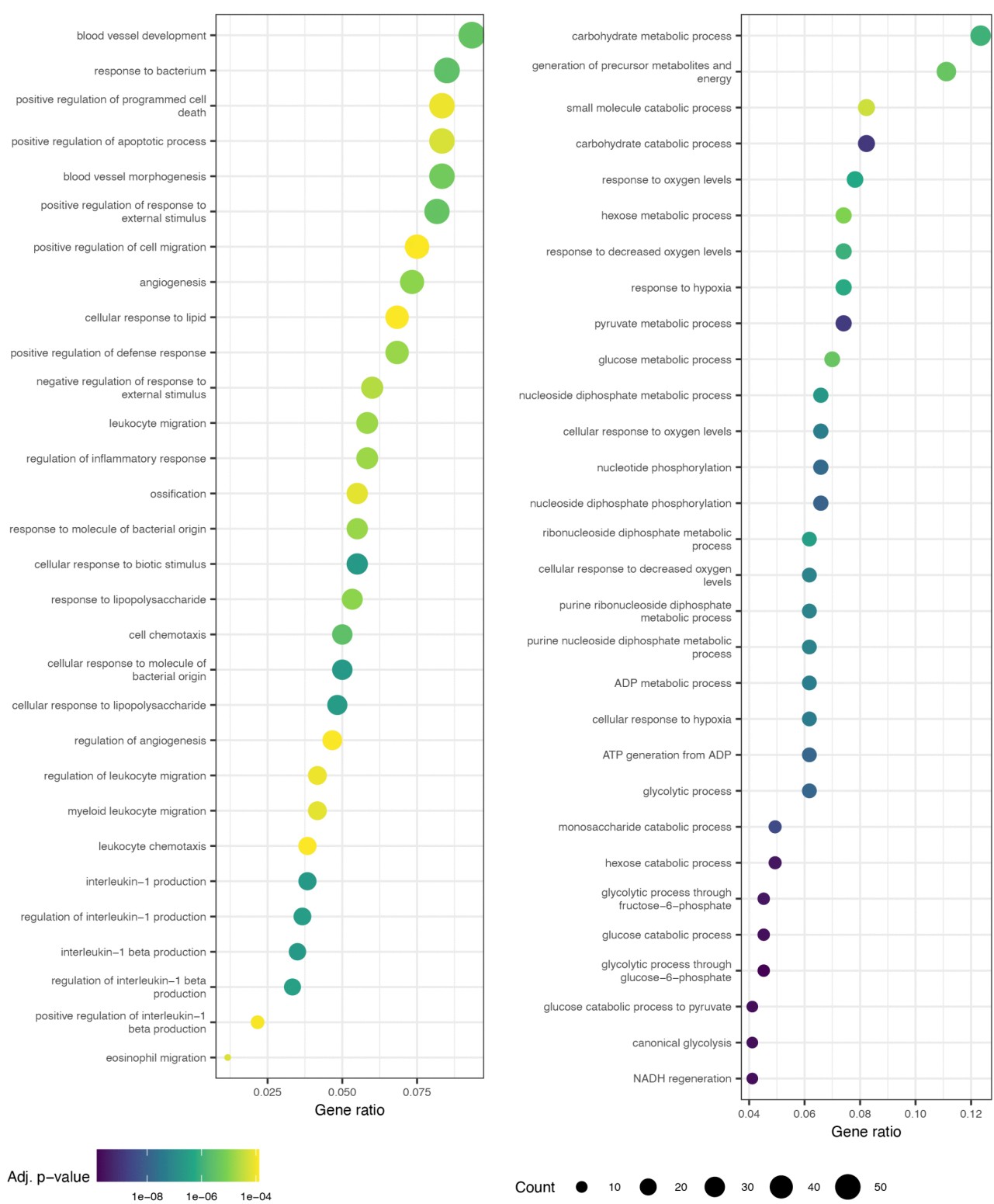

**Fig 8. GO terms enriched for genes differentially expressed between 0h and 1h or 8h and 12h GCSF samples.** GO terms enriched for DEGs between 0h and 1h and between 8h and 12h are shown on the left and right respectively. See the legend of Fig 7 for plot description.

## Inference of gene regulatory networks

We inferred putative gene regulatory networks (GRNs) by identifying TFs expressed in correlated temporal patterns. We computed the Pearson's correlation coefficient between the expression patterns of each pair of TF and used it as a similarity metric to cluster the TFs. First, we focused on TFs previously implicated in macrophage-neutrophil differentiation [3,7,4]. The TFs separate into three distinct groups with those specifying the same lineage clustering together (S36 Fig). The first group, comprising Gfi1, Cebpa, Cebpe, Per3, corresponds to known neutrophil TFs, the second, Gata2 and Sox4, to stem/progenitor TFs, and the last, Irf8, Egr1/2, Zeb2, and Nab2, to macrophage TFs. Annotating the TFs by their dominant—highest weight—behavior reveals that all the neutrophils TFs exhibit behavior 2, so that they are transiently upregulated in GCSF condition with expression peaking around 100h. The macrophage TFs, Irf8, Egr1, and Egr2 are dominated by behavior 9, transiently expressed from 12h to 128h, whereas Nab2 and Zeb2 are dominated by behavior 10, upregulated after 80h and peaking at the end of the differentiation. The strong correspondence of the GRNs inferred from the correlation of TF temporal expression patterns to their known functions determined in BM implies that PUER differentiation is guided by substantially similar GRNs as in BM. Furthermore, the later expression of Nab2 and Zeb2 relative to Irf8, Egr1, and Egr2 implies that the former TFs act downstream of the latter, in agreement with previous work [3,4].

Hierarchical clustering of 2,331 mouse TFs and cofactors (TFCheckpoint2.0) [26] shows at least 6 modules of co-expression (S37 Fig). Three of the six modules contain previously implicated TFs, implying that these TFs are embedded in much larger GRNs for the macrophage and neutrophil fates. There are also three modules that do not contain any core TF and could thus potentially be novel GRNs. Reflecting the hierarchical clustering of all transcripts (S17 Fig), most of the TFs in each module have the same dominant behavior and grouping them by dominant behavior alone results in significant block structure and highly distinct expression patterns (S38 Fig). The high level of correlation between the TFs having the same dominant behavior and the embedding of the known core TFs amongst them suggests that TFs sharing the same dominant behavior are interconnected in GRNs that orchestrate the expression patterns of the rest of the behavior's genes.

## Discussion

Time series datasets from differentiation experiments [9] are a powerful method for inferring GRNs and clarifying the causality of gene regulatory events during development [27]. While such data are acquired *in vitro* and may be only an approximation to the *in vivo* phenomena, they provide a window into dynamics that is not available in static snapshots of developmental processes. While pseudotime approaches [16,20] have been used to infer the developmental sequence of cells in scRNA-Seq data, it is impossible to determine the rate of change of expression with time. Furthermore, *in vitro* differentiation allows one to conduct carefully controlled experiments to minimize the effect of animal-to-animal and technical variation. Despite these strengths, high-resolution time series datasets are fairly uncommon and we are only aware of one other dataset [9] of comparable temporal resolution that has been published so far in hematopoiesis.

There is a long and fruitful history of utilizing inducible cell lines such as PUER [3,7,15], G1ME [28], FDCP [9,29–31], and others to gain a mechanistic understanding of hematopoiesis. The data collected in this study offered an opportunity to systematically compare PUER cells and their differentiation to *in vivo* BM differentiation in terms of the global transcriptomic state. After batch correction, differentiated PUER cells were closer in their state to BM cells than to other PUER cells save the closest time points, BM cells being among their ten nearest neighbors. Undifferentiated PUER cells were the most distant from any BM cells, which is consistent with their mutant *Spi1*$^{-/-}$ genotype. Later timepoints of GCSF-OHT or IL3-OHT cells were closest to BM neutrophils or monocytes respectively. This showed that PU.1 activation by OHT treatment rescues *Spi1*$^{-/-}$ PUER cells and reprograms them towards *in vivo* BM neutrophils or monocytes. The patterns of the evolution of the global transcriptomic state were, by and large, reflected in the temporal expression patterns of known marker and TF genes (S6 Fig–S11 Fig) and the known regulatory relationships within cell-fate GRNs appear to be preserved in PUER cells (S36 Fig). These observations, taken together with the demonstration that PUER

cells remain bipotential until OHT treatment (S12 Fig–S13 Fig), show that PUER is a model system for bipotential lineage decisions.

Our analyses suggest that there is a large-scale sudden change in gene expression, reminiscent of phase transitions, occurring around 8 – 12h after the induction of PU.1 by OHT. We confirmed this conclusion in multiple different analyses: the Pearson correlation of genome-wide gene expression (Fig 2A and 2B), PCA (Fig 2C), and expression patterns of differentially expressed genes (S14 Fig). Furthermore there is another transition occurring around 80h. Similar to our data, it has been observed that there is relatively low rate of change between 12h and 48 hours and from 72h to 168h in the reprogramming of B cells into macrophages by the enforced expression of *Cebpa* [32]. This suggests perhaps that these transitions are a general phenomenon of reprogramming and not an idiosyncrasy of the PUER system. It wasn't clear whether the jumps between 0h and 12h and 48h and 72h during B cell transdifferentiation were the result of the relatively large time intervals, 12h and 24h, or a significantly higher velocity of gene expression change. The much higher temporal resolution of our data unambiguously establishes that the jumps are the result of increased velocity—the shift between 8h and 12h in IL3 conditions is the second largest shift after the one induced by GCSF pre-treatment but occurs in 4h instead of 48h (Fig 2C).

The trajectories of the DEGs between the endpoints of differentiation suggested extensive transient regulation so that the change in expression from undifferentiated cells peaks during the course of differentiation rather than at the end (S14 Fig). We also observed that each gene was not expressed in a unique temporal pattern; instead there were a relatively small number of patterns in each of which thousands of genes were expressed. These observations motivated two questions. First, are there genes that are not detectable as DEGs between the endpoints of differentiation but are nevertheless transiently regulated? An affirmative answer to this question would expand the set of genes participating in the information transfer process during differentiation. Second, what number and type of temporal expression patterns are sufficient to describe the changes in gene expression genome wide? Dimensionality reduction would make it easier to appreciate the larger scale patterns during differentiation.

NMF revealed that only 7–10 patterns or behaviors were sufficient to recapitulate genome-wide expression with high fidelity. To a large extent, individual transcripts exhibited these behaviors (Fig 5) so that the behaviors reflect actual regulatory events during differentiation. Confirming our suspicions, approximately 15,000 transcripts have transient expression whereas only about 8,000 are differentially expressed between endpoints (Fig 3). The transient expression patterns are of a specific type; OHT treatment causes thousands of genes to be expressed in a pulsatile fashion with varying pulse initiation times and durations (Table 1). This implies that information transfer during differentiation occurs in coordinated waves of gene expression culminating in the permanent turning on of certain genes after ~80h. Furthermore, the time span of the pulses matches the sharp transitions observed in the transcriptome-wide data (Fig 2). The 8 – 12h transition coincides with the termination of the behavior 6 pulse and the initiation of the behavior 8 and 9 pulses. The transition at 80h corresponds to the termination of behavior 8 and 9 pulses and the initiation of behavior 4 and 10 that remain upregulated until the end.

GO analysis of the genes with high weights for each behavior showed that most of the behaviors specialized in specific physiological processes (Table 1 and S26 Fig–S35 Fig). This implies that physiological remodeling proceeds in a characteristic order during differentiation. The earliest event is the transient upregulation of signal transduction pathways (behavior 3 and 6) during the first 8 hours followed by translation and mRNA processing (behavior 7) from 2h to 48h, and metabolism (behaviors 8 and 9) from 8h to 128h. The last set of genes to be upregulated around 80 – 100h are enriched for myeloid phenotypic processes such as neutrophil chemotaxis and innate immune response (behaviors 4 and 10). The GO analysis of the behaviors is corroborated by the pairwise comparisons between early timepoints (Fig 7, Fig 8, S22 Fig–S25 Fig). GO terms related to signal transduction and leukocyte differentiation are enriched among the differentially expressed genes in the comparisons between consecutive time points up to 2h. However, in the comparisons between 4h and 8h and 8h and 12h, signal transduction or myeloid differentiation terms are no longer in the top 30. Taken together

these observations imply that the first transition at 8h corresponds to the completion of the remodeling of signal transduction, differentiation, transcription, and translation and the initiation of metabolic rewiring. The second transition around 80h is the completion or slowing down of metabolic remodeling and the upregulation of genes producing the phenotypic properties of the cell type. A corollary of this conclusion is that the cell-fate decision appears to have been made by the time of the $8 - 12h$ transition; subsequent modulation of gene expression revolves around phenotypic remodeling.

NMF was also successful in clustering genes in a biologically meaningful manner in an independent and very different dataset from the one analyzed here. NMF inferred behaviors that corresponded to lineages and developmental stages in Tusi *et al.*'s BM scRNA-Seq dataset [16] without being provided any prior information about cell types or markers. We speculate that this is because genes are regulated in space and time in an additive manner by enhancers that drive spatiotemporally restricted expression patterns [33]. NMF, by decomposing the expression of each gene as a sum of behaviors, models this additive process. Genes expressed in two different cell types are members of two groups, one for each cell type. In contrast, hierarchical clustering would group genes expressed in two different cell types as a novel third cluster. NMF also performed better than hierarchical clustering in identifying groups of genes expressed in a pulsatile manner at early timepoints. This could be because such genes, being expressed at few timepoints, would have low correlation with most genes and may get clustered in an unpredictable manner. Lastly, while our current implementation of NMF does not assume any particular form of time dependence of the behaviors beforehand, it would be possible to constrain the behaviors to satisfy particular ones during optimization in the future to test hypotheses about temporal expression patterns. Overall, our experience with NMF suggests that it a relatively simple and computationally efficient way to uncover spatiotemporal gene expression programs in developmental contexts.

This dataset is a rich resource and the analysis reported here scratches the surface of the biological insights harbored within. One of the main goals of future work would be to clarify the causality of events at a finer granularity both in time and at the level of individual genes. A potential way of accomplishing that would be to conduct "multi-omic" analyses by combining these data with epigenetic assays such as ATAC-Seq [34,35]. We [35] and others [34] have demonstrated that TF binding sites are detectable at single base pair resolution in deeply sequenced ATAC-Seq data. Therefore, it is possible in principle to determine the TFs controlling each behavior by inspecting the promoters and enhancers of member genes for TF footprints. This could result in a "blow-by-blow" description of the causality of the information transfer process during cellular differentiation. Finally, the NMF analysis, besides illuminating the information transfer process, has remarkably demonstrated that 36,255 transcripts effectively behave as 7–10 metagenes. This dimensionality reduction could enable future predictive models of macrophage-neutrophil differentiation or of multiple lineages, if combined with other datasets [9,16].

## Methods

### PUER cell culture

PUER cells were cultured according to standard procedures [14]. PUER cells were routinely maintained in complete Iscove's Modified Dulbecco's Medium (IMDM; Gibco, 12440061) supplemented with 10% FBS, $50 \mu M$ $\beta$-mercaptoethanol and 5ng/mL IL3 (Peprotech, 213–13).

### PUER differentiation

PUER cells were expanded in T-75 flasks prior to the initiation of differentiation. GCSF pre-treatment was initiated by washing the cells 3 times with PBS and then seeding in 48-well plates at a concentration of $5 \times 10^5$ cells/ml in PUER cell culture medium in which IL3 had been replaced by 10 ng/mL of Granulocyte Colony Stimulating Factor (GCSF; Peprotech, 300–23). At the same time, the PUER cells destined for the macrophage differentiation were seeded in 48-well plates at a concentration of $5 \times 10^5$ cells/ml in IL3-conditioned PUER cell culture medium. After 48 hours of GCSF pre-treatment, the

neutrophil differentiation was commenced by the addition of 100 nM of 4-hydroxy-tamoxifen (OHT; Sigma, H7904-5MG) to the GCSF-conditioned media. The macrophage differentiation was initiated at the same time by the addition of 200 nM of OHT. We regard time zero (0h) of differentiation as occurring just before the addition of OHT. With this starting point, the initiation of GCSF pre-treatment occurs immediately after −48h while cells pre-treated with GCSF for 48h but not yet induced by OHT are at 0h. Since both treatments start with uninduced PUER cells, the data for the −48h time point of the neutrophil differentiation and 0h time point of the macrophage differentiation are derived from the same samples and are identical. OHT rapidly converts from the Z isomer to the E isomer, having 100-fold lower activity, in cell culture media. Half the medium was replaced, and fresh OHT was added at 40h, 88h, and 136h in order to maintain differentiation pressure.

## Switch experiment

In the switch experiment, cells were seeded in GCSF as described above. After 48 hours of GCSF pre-treatment, the cells were washed in IL3 medium and then seeded in IL3 medium. Macrophage differentiation was initiated by adding 200nM OHT. IL3-OHT, GCSF-OHT, or switched cells were harvested at 96h. The cells' morphology was assessed by staining with Wright Geimsa modified (Sigma, WG16) as described previously [7]. Cell-surface differentiation markers were analyzed by flow cytometry on a CytoFLEX S (Beckman Coulter) instrument as described previously [14]. The cells were stained with the Ghost Dye Blue 516 live-dead stain (Cytek, 13–0867-T100) and Cd117-PE (ACK2; Cytek 50–1172-U025), Cd11b-PECy7 (M1/70; Cytek 60–0112-U025), and Cx3cr1-SuperBright 702 (2A9-1; Thermo Fisher 67-6099-42) antibodies.

## Sample collection

In addition to undifferentiated PUER cells, which correspond to the −48h neutrophil and 0h macrophage timepoints, samples were collected after 48h of GCSF pre-treatment but before OHT induction (0h neutrophil). After OHT induction, in both GCSF and IL3 treatments, samples were collected every hour for the first four hours, every four hours for the first day, and every eight hours until the end of the seventh day (Fig 1A). 4 biological replicates were collected for each time-point. Cells for each time point were seeded into a dedicated 48-well plate so that they could be harvested without disturbing the samples for the other time points. Since the differentiation produces adherent cells, the cells were detached with trypsin using standard protocols for all timepoints after 24h. The cells were transferred into a 96-well plate, which was centrifuged at 1500rpm for 5 min, the majority of the medium was aspirated, the cell pellet was snap-frozen in liquid nitrogen, and stored at −80C until RNA extraction.

## Total RNA extraction, quality control, and spike in of ERCC standards

Total RNA was extracted on a Bio-Mek FX$^P$ liquid handling workstation (Beckman Coulter) using the RNAdvance Tissue total RNA isolation kit (Beckman Coulter, A32649) from 3 replicates in a 96-well format following the manufacturer's protocol. Genomic DNA contamination was assessed by reverse transcribing the RNA with and without reverse-transcriptase and detecting GAPDH using qPCR. The number of additional cycles required to reach a threshold ($\Delta C_t$) was utilized to assess the fraction of genomic DNA ($2^{-\Delta C_t}$) and only samples with less than 1% genomic DNA were utilized for library preparation. The quality of the RNA was assessed by capillary gel electrophoresis on the Agilent 2100 Bioanalyzer using the Eukaryote Total RNA Nano kit (Agilent, 5067–1511). Only samples with RNA integrity numbers (RIN) greater than or equal to 9.5 were utilized, although there was only one sample with a RIN of 9.5 and the median RIN was 9.9. The concentration of RNA was determined using the Qubit fluorometer and the Qubit RNA High Sensitivity kit (Invitrogen, Q32855). With the exception of 8 samples with lower RNA yield, the samples were standardized to a mass of 1,875 $ng$ in a $25\mu l$ volume. The samples with lower yield were standardized to a mass of 1,437.5 $ng$ in a $25\mu l$ volume. $3.75\mu l$ or $2.88\mu l$ of a 1:100 dilution of the External RNA Control Consortium (ERCC) ExFold RNA Spike-In mix (Invitrogen, 4456739) was added to the high- and low-yielding samples respectively.

## CUT&RUN

The CUT&RUN assay was performed using the CUTANA CUT&RUN kit (Epycypher, 14–1048), following the manufacturer's instructions. Briefly, $1 \times 10^5$ undifferentiated and 48h IL3-OHT PUER cells were harvested, immobilized to activated Concanavalin A beads and permeabilized with digitonin. $0.5\mu$g of $\alpha$-PU.1 (Santa Cruz, sc-390405) primary and a 1:200 dilution of Rabbit anti-Mouse IgG (Cytek, 70–8076-M002) secondary were added to the samples before treatment with pAG-MNase for 2 hours. The released DNA fragments were purified and libraries prepared using the CUTANA CUT&RUN Library Prep Kit (Epicypher, 14–1001). The CUT&RUN libraries were sequenced to an approximate depth of $5 \times 10^6$ paired-end 150 bp reads per sample.

## Library preparation and RNA sequencing

Illumina libraries were prepared by Novogene Corporation Inc. (Chula Vista, CA) using the NEB Ultra II RNA Library Prep Kit for Illumina according to manufacturer protocols. The libraries were sequenced on an Illumina Novaseq 6000 S2 2 × 150 bp flow cell to an average depth of $29.24 \times 10^6$ raw reads per sample for a total of $4.82 \times 10^9$ reads. The sequencing provider filtered the reads to remove ones containing Illumina adaptors, or more than 10% indeterminate bases ("N"s), or having more than half bases with a phred score below 5. The remaining "clean" reads, representing 95.52% of raw reads, were processed further as described below.

## RNA-Seq data processing

We used Salmon [36] to map the reads to the GRCm38 reference genome with the selective alignment strategy and to determine transcript abundances. The median-of-ratios method [37,38] of DESeq2, which computes the ratio of each transcript's abundance in each sample to the geometric mean of the expression across samples and then determines a size factor for each sample as the median of these ratios, was used to normalize gene expression to library size.

## CUT&RUN data processing

The quality of the reads was assessed using FASTQC (v0.12.1) and trimming was performed with Trimmomatic (v0.39) [39]. Reads were aligned to the mm10 reference using Hisat2 (v2.2.0) [40]. Duplicates were marked and removed using Picard. PU.1 footprints were enriched by filtering out inserts longer than 120 bp. Peaks were called using MACS2 (v2.2.9.1) [41] with an FDR of 0.05.

## Principal component analysis

Principal component analysis (PCA) was carried out with the `prcomp` R function with the `scale` parameter set to `TRUE`. Transcripts were filtered to remove those having a normalized count less than 5 in every sample—hereafter referred to as low-expressed transcripts.

## Hierarchical clustering and gene expression heatmaps

Genes were clustered using the `hclust` function of the `stats` **R** package and visualized using either `pheatmap` function or the `Heatmap` function of the `ComplexHeatmap` package [42]. Samples or timepoints were clustered hierarchically, after excluding the low-expressed transcripts, using Pearson correlation, $r$, as a similarity metric; the distance was computed as $1 - r$. TF co-expression analysis was perfomed on a set of 2,331 annotated TFs and co-factors (TFCheckpoint2.0 [26]).

## Outlier detection

Principal components analysis (PCA) was performed on the filtered data and the $z$-score of the $k$th principal component of each replicate $i$ at a given time point was calculated as

$$z_i^k = \left| \frac{PC_i^k - \frac{1}{N}\sum_{i=1}^{N} PC_i^k}{\frac{1}{N}\sum_{i=1}^{N}\left(PC_i^k - \frac{1}{N}\sum_{i=1}^{N} PC_i^k\right)^2} \right|,$$

where $k = 1,2,3,4$ are the top four PCs accounting for 96% of the total variation in the data and $PC_i^k$ is the principal component score of the $k$th component. A replicate was regarded as an outlier if $z_i^k > 2$ for any of the top four PCs. No outliers were detected in the data using this method.

In a complementary approach to identifying outliers we clustered all the samples hierarchically (S1 and S2 Figs). All three replicates for the 96h and 144h timepoints in IL3 conditions appeared to be visually significantly different from the time points immediately preceding or succeeding them. The time points immediately before and after the outliers were more correlated to each other than to the outliers (S3 Fig). These timepoints were unusual, having been preceded by supplementation with OHT (see above), so that the differences could be transient effects of OHT. We excluded these two timepoints from further analyses.

## Differential gene expression analysis

Differential expression analysis was conducted using DESeq2 [43], which uses the Wald test to determine the significance of the estimated fold change and the Benjamini-Hochberg procedure to correct for multiple testing. A transcript was considered differentially expressed if its adjusted $p$-value was less than 0.05 and the $\log_2$ fold change (FC) was more than 0.58 (±50% change).

## Non-negative matrix factorization

Non-negative matrix factorization (NMF) [24,25] was used to extract features, reduce dimensionality, and cluster the temporal expression patterns of all the transcripts in the dataset. Prior to NMF, the expression of each gene across all time-points and conditions was scaled so that maximum expression of each gene is 1.

## NMF Implementation

The algorithm produces a lower-dimensional representation of the dataset by approximating the temporal pattern of each gene's expression as a linear sum of a few characteristic temporal patterns, referred to here as *behaviors*. The algorithm does so by approximately factoring the gene expression matrix **X** into two lower-dimensional non-negative matrices, **W** and **H**, as

$$X_{gt} \approx (\mathbf{WH})_{gt} = \sum_{m=1}^{M} W_{gm}H_{mt} \quad \forall g, t. \tag{1}$$

Here $g = 1, \ldots, G$, $t = 1, \ldots, T_G, T_G + 1, \ldots, T_G + T_I$, and $m = 1, \ldots, M$ index genes, timepoints, and behaviors respectively. $G$ and $M$ are the total number of genes and behaviors respectively. $T_G$ and $T_I$ are the total number of timepoints in the GCSF and IL3 conditions respectively. $X_{gt}$ is the expression of gene $g$ at timepoint $t$. $H_{mt}$ is the expression of behavior $m$ and timepoint $t$. $W_{gm}$ is the weight of behavior $m$ in the linear combination of behaviors that approximates the expression pattern of gene $g$. The expression of behavior $m$ at a given time point $t$, $H_{mt}$, is independent of its expression at any other time point and therefore NMF does not assume any particular form of time dependence beforehand. Instead, the algorithm allows the requirement of approximating $X_{gt}$ implicitly infer which forms of time dependence best describe the empirical observations.

The algorithm iteratively updates the **W** and **H** matrices to converge to a local maximum of the objective function $O$ given by

$$O(\mathbf{W}, \mathbf{H}) = \sum_{g=1}^{G} \sum_{t=1}^{T} \left( X_{gt} \log \left( \mathbf{WH} \right)_{gt} - \left( \mathbf{WH} \right)_{gt} \right).$$

(2)

The alternating gradient method was used to produce a sequence $\{\mathbf{W}^k, \mathbf{H}^k\}$, where $k = 1, \ldots, N$, by applying the multiplicative update rules

$$\mathbf{W}^{k+1} = \mathbf{W}^k \circ \frac{\mathbf{X} \left( \mathbf{H}^k \right)^T}{\mathbf{W}^k \mathbf{H}^k \left( \mathbf{H}^k \right)^T}$$

(3)

and

$$\mathbf{H}^{k+1} = \mathbf{H}^k \circ \frac{\left( \mathbf{W}^k \right)^T \mathbf{X}}{\left( \mathbf{W}^k \right)^T \mathbf{W}^k \mathbf{H}^k},$$

(4)

where ∘ signifies element-wise multiplication. The elements of the initial matrices $\mathbf{W}^0$ and $\mathbf{H}^0$ were populated by sampling from the uniform distribution over $[0,1)$

We calculated the Frobenius norm ($F^k$) of the difference between original $\mathbf{X}$ matrix and its $k$th approximation, $\overline{\mathbf{X}}^k = \mathbf{W}^k \mathbf{H}^k$,

$$F^k = \left\| \mathbf{X} - \overline{\mathbf{X}}^k \right\|.$$

(5)

$\mathbf{W}^k$ and $\mathbf{H}^k$ were updated according to Equations 3 and 4 until the stopping criterion, that the absolute change in the Frobenius norm between successive iterations was less than the threshold $\theta$,

$$|F^k - F^{k-1}| < \theta,$$

(6)

was met. Unless specified otherwise, $\theta$ was chosen to be 0.05.

**Choice of the number of behaviors $M$**

We found that, as the number of metagenes M was increased, the root mean square error (RMSE) between successive $\overline{\mathbf{X}}$ approximations decreased to a minimum around $M=7$ (S4 Fig). The RMSE was computed as $\frac{F}{G \times M}$, where $F$ is the Frobenius norm of the difference of successive $\overline{\mathbf{X}}$ approximations. This suggested that NMF, viewed as an unsupervised learning algorithm, was able to satisfactorily "learn" the patterns in the data with $M=7$, and that going too far beyond this point might be redundant and cause overfitting. We chose $M=10$ to ensure that the full diversity of temporal expression patterns was accounted for.

**Robustness analysis**

The robustness of the NMF algorithm to random initial conditions was checked by repeating the factorization 100 times and comparing the RMSE between each pair of approximated gene expression matrices to the RMSE between the approximated and actual gene expression matrices (S5 Fig). The RMSE between different approximations was found to be consistently lower than the RMSE between the approximated and actual gene expression matrices, demonstrating that the approximate solutions are robust to the random initialization of the NMF algorithm.

## Visualization

The visualizations in Fig 5 were produced in two steps. In the first step, the behaviors were sorted according to the condition and time at which their expression peaked so that behaviors peaking earliest in GCSF conditions had the lowest rank, followed by those expressed later in GCSF, then by those expressed early in IL3, and finally by those expressed late in IL3. In the second step, for each gene, the behavior with the highest weight—the dominant behavior—was identified. The genes were grouped according to their dominant behavior and then sorted in descending order of the weight of the dominant behavior.

## Gene ontology analysis

`clusterProfiler` [44] was used to identify over-represented biological process Gene Ontology (GO) terms associated with differentially expressed genes (DEGs) [45] or the top 500 genes associated with a behavior, and having a maximum normalized expression of at least 3, using /all annotated genes as the background gene list. A functional annotation category was considered significant if the Benjamini-Hochberg adjusted $p$-value was less than 0.05.

## PU.1 enrichment analysis

PU.1 peaks from 48h IL3-OHT differentiated samples were used to identify genes that bind PU.1 either at or near gene transcription start sites (TSSs; ±1kb from TSS) or in distal regions ($\pm 50kb$ from TSS). A transcript's start site is given by the genomic location of its first base. For genes expressing multiple transcripts, only the one driving the highest expression in undifferentiated PUER cells was chosen for analysis.

To compute the enrichment of PU.1 binding among DEGs between consecutive timepoints, we determined the ratio of the fraction of DEGs bound by PU.1 to the fraction of genes bound by PU.1 in the background set. Enrichment was calculated as $\frac{T_p/T_n}{A_p/A_n}$, where $T_p$ and $A_p$ are the number of genes bound by PU.1 in the DEG and background sets respectively while $T_n$ and $A_n$ are the number of genes in the DEG and background sets respectively. The background was the set of all genes having normalized expression of 5 or more. DEGs from the 0h-1h, 1h-2h, 2h-3h, 3h-4h, 4h-8h, 8h-12h, 72h-80h, 128h-136h, and 160h-168h comparisons in GCSF or IL3 conditions were scanned for PU.1 peaks both within ±1kb and $\pm 50kb$ excluding the ±1kb region using the `subsetByOverlaps()` function in the `IRanges` **R** package [46]. The process was repeated for all genes in the background set. The enrichment of PU.1 binding in the top 500 genes of each behavior was computed in a similar manner. The significance of the enrichment ratio was calculated using the one-sided Fisher's exact test.

## Supporting information

**S1 Fig. Genome-wide gene expression in all IL3 samples.** Color indicates the expression of each transcript ($y$-axis) in each IL3 sample ($x$-axis) with maximum expression scaled to 1. Transcripts were clustered hierarchically using Pearson correlation as a similarity measure.
(PDF)

**S2 Fig. Genome-wide gene expression in all GCSF samples.** Color indicates the expression of each transcript ($y$-axis) in each GCSF sample ($x$-axis) with maximum expression scaled to 1. Transcripts were clustered hierarchically using Pearson correlation as a similarity measure.
(PDF)

**S3 Fig. Correlation in gene expression between selected samples.** Correlation in gene expression between suspected outliers and immediate neighbors (88h vs 96h, 96h vs 104h, 136h vs 144h, and 144h vs 152h). Correlation between the timepoint immediately preceding and the timepoint immediately following the suspected outlier (88h vs 104h and 136h vs 152h).
(PDF)

**S4 Fig. Choice of the number of metagenes in the NMF approximation.** The root mean square error (RMSE) between the gene expression matrix and its NMF approximation (orange crosses) or the RMSE between successive approximations (blue dots) as the number of metagenes ($M$) is varied.
(PDF)

**S5 Fig. Robustness of NMF.** Histograms of the RMSE between the gene expression matrix and 100 NMF approximations (orange) or the RMSE between the approximations (blue). Dashed lines are the Gaussian densities with the mean and standard deviation of each histogram.
(PDF)

**S6 Fig. Comparison of HSC marker gene expression in PUER cells to BM. A**. SPRING plot of BM cells. Cells are colored according to the probability of differentiation computed by *Tusi et al.* [16], which is inversely related to the probability of a cell being an HSC. **B-D**. Gene expression of HSC markers during PUER differentiation (left) and in BM (right; SPRING plot). Errors bars indicate standard deviation.
(PDF)

**S7 Fig. Comparison of HSC TF gene expression in PUER cells to BM. A**. SPRING plot of BM cells. Cells are colored according to the probability of differentiation computed by *Tusi et al.* [16], which is inversely related to the probability of a cell being an HSC. **B-F**. Gene expression of HSC TFs during PUER differentiation (left) and in BM (right; SPRING plot). Errors bars indicate standard deviation.
(PDF)

**S8 Fig. Comparison of neutrophil marker gene expression in PUER cells to BM. A**. SPRING plot of BM cells. Cells are colored according to the PBA probability of being a neutrophil. **B-L**. Gene expression of neutrophils markers during PUER differentiation (left) and in BM (right; SPRING plot). Errors bars indicate standard deviation.
(PDF)

**S9 Fig. Comparison of neutrophil TF expression in PUER cells to BM. A**. SPRING plot of BM cells. Cells are colored according to the PBA probability of being a neutrophil. **B-E**. Gene expression of neutrophil TFs during PUER differentiation (left) and in BM (right; SPRING plot). Errors bars indicate standard deviation.
(PDF)

**S10 Fig. Comparison of macrophage marker expression in PUER cells to BM. A**. SPRING plot of BM cells. Cells are colored according to the PBA probability of being a macrophage. **B-I**. Gene expression of macrophage markers during PUER differentiation (left) and in BM (right; SPRING plot). Errors bars indicate standard deviation.
(PDF)

**S11 Fig. Comparison of macrophage TF expression in PUER cells to BM. A**. SPRING plot of BM cells. Cells are colored according to the PBA probability of being a macrophage. **B-G**. Gene expression of macrophage TFs during PUER differentiation (left) and in BM (right; SPRING plot). Errors bars indicate standard deviation.
(PDF)

**S12 Fig. PUER cell morphology in the switch experiment. A-D.** PUER cells stained with Wright Giemsa. **A**. Undifferentiated cells **B**. 96h IL3-OHT (macrophage). **C**. 96h GCSF-OHT (neutrophil). **D**. "Switch". Cells pre-treated with GCSF for 48 hours and then switched to IL3 prior to OHT addition. Cells were treated with OHT for 96 hours. **E**. The distribution of cell morphology in each treatment. Early or undifferentiated cell types include myeloblasts, monoblasts, and promyelocytes [14,47]. Neutrophils include myelocytes, metamyelocytes, band cells, and segmented neutrophils. Monocytes

include both mature monocytes and macrophages. Error bars are $\pm\sigma$, where $\sigma$ is the standard deviation computed over two (undifferentiated, switch) or three (96h IL3-OHT, 96h GCSF-OHT) replicates.
(PDF)

**S13 Fig. The expression of cell-surface lineage markers in the switch experiment.** Dot plots of Cd117, Cd11b, and Cx3cr1 expression in Undifferentiated, 96h IL3-OHT (macrophage), 96h GCSF-OHT (neutrophil) and "switch" samples. In the switch experiment, cells were pre-treated with GCSF for 48 hours and then switched to IL3 prior to OHT addition. Cells were treated with OHT for 96 hours.
(PDF)

**S14 Fig. Transient expression of differentially expressed genes.** The expression of all differentially expressed genes, with maximum expression scaled to 1, is shown as a color map. Genes are clustered hierarchically according to the Pearson correlation of scaled temporal expression patterns. (A) GCSF. Plotted genes are the union of DEGs identified in the comparisons between the $-48h$ and $168h$, $-48h$ and $0h$, and $0h$ and $168h$ GCSF samples (Fig 3A–C). (B) IL3. DEGs identified in the comparison between the $0h$ and $168h$ IL3 samples (Fig 3D).
(PDF)

**S15 Fig. The temporal behaviors of the 10 metagenes identified by NMF when applied to unscaled data.** The temporal behavior of each metagene is shown in red. The temporal expression patterns of the transcripts having the highest 200 weights for each behavior are shown in black.
(PDF)

**S16 Fig. Clustering of all expressed transcripts by dominant behavior.** The similarity between the temporal expression of each pair of genes, as measured by the distance $1-r$, where $r$ is Pearson's correlation coefficient is plotted as a heat map ("distance matrix"). The genes are ordered by their dominant behavior, annotated on the left. The center and right heatmaps show the temporal expression pattern of each genes.
(TIFF)

**S17 Fig. Hierarchical clustering of all expressed genes.** Genes are hierarchically clustered. See the legend of S16 Fig for the plot description.
(TIFF)

**S18 Fig. The behaviors of the 17 metagenes identified by applying NMF to Tusi *et al.*'s Kit + BM progenitor single-cell gene expression data.** The behavior in each cell is shown as a SPRING plot. The SPRING plot at the bottom right indicates which cells have greater than 80% PBA probability of belonging to each lineage. Gene expression was scaled so that the maximum for each gene is 1.
(TIFF)

**S19 Fig. The behaviors of the 17 metagenes identified by applying NMF to Tusi *et al.*'s Kit + BM progenitor single-cell gene expression data without prior scaling.** See the legend of S18 Fig for description.
(TIFF)

**S20 Fig. The enrichment of PU.1 binding to regions neighboring genes differentially expressed between consecutive timepoints.** The enrichment of PU.1 binding to regions neighboring the DEGs between consecutive timepoints relative to the background set is plotted on the $y$-axis. The enrichment is plotted separately for all DEGs ("all"), upregulated DEGs ("up"), or downregulated DEGs ("down") and for binding to proximal (±1kb) or distal (±50kb) regions.
(PDF)

**S21 Fig. The enrichment of PU.1 binding to regions neighboring the top 500 genes for each behavior.** The enrichment of PU.1 binding to regions neighboring the top 500 genes for a behavior relative to the background set is plotted on the *y*-axis. The red line indicates no enrichment.
(PDF)

**S22 Fig. GO terms enriched for genes differentially expressed between 1h and 2h and 2h and 3h IL3 samples.** GO terms enriched for DEGs between 1h and 2h and between 2h and 3h are shown on the left and right respectively. See the legend of Fig 7 for plot description.
(PDF)

**S23 Fig. GO terms enriched for genes differentially expressed between 1h and 2h and 2h and 3h GCSF samples.** GO terms enriched for DEGs between 1h and 2h and between 2h and 3h are shown on the left and right respectively. See the legend of Fig 7 for plot description.
(PDF)

**S24 Fig. GO terms enriched for genes differentially expressed between 3h and 4h and 4h and 8h IL3 samples.** GO terms enriched for DEGs between 3h and 4h and between 4h and 8h are shown on the left and right respectively. See the legend of Fig 7 for plot description.
(PDF)

**S25 Fig. GO terms enriched for genes differentially expressed between 3h and 4h and 4h and 8h GCSF samples.** GO terms enriched for DEGs between 3h and 4h and between 4h and 8h are shown on the left and right respectively. See the legend of Fig 7 for plot description.
(PDF)

**S26 Fig. GO terms enriched for the top 500 behavior 1 genes.** See the legend of Fig 7 for plot description.
(PDF)

**S27 Fig. GO terms enriched for the top 500 behavior 2 genes.** See the legend of Fig 7 for plot description.
(PDF)

**S28 Fig. GO terms enriched for the top 500 behavior 3 genes.** See the legend of Fig 7 for plot description.
(PDF)

**S29 Fig. GO terms enriched for the top 500 behavior 4 genes.** See the legend of Fig 7 for plot description.
(PDF)

**S30 Fig. GO terms enriched for the top 500 behavior 5 genes.** See the legend of Fig 7 for plot description.
(PDF)

**S31 Fig. GO terms enriched for the top 500 behavior 6 genes.** See the legend of Fig 7 for plot description.
(PDF)

**S32 Fig. GO terms enriched for the top 500 behavior 7 genes.** See the legend of Fig 7 for plot description.
(PDF)

**S33 Fig. GO terms enriched for the top 500 behavior 8 genes.** See the legend of Fig 7 for plot description.
(PDF)

**S34 Fig. GO terms enriched for the top 500 behavior 9 genes.** See the legend of Fig 7 for plot description.
(PDF)

**S35 Fig. GO terms enriched for the top 500 behavior 10 genes.** See the legend of Fig 7 for plot description.
(PDF)

**S36 Fig. Hierarchical clustering of key cell-fate TFs by temporal expression.** The similarity between the temporal expression of each pair of TFs, as measured by the distance $1-r$, where $r$ is Pearson's correlation coefficient is plotted as a heat map. Genes have been annotated by their behavior on the left.
(PDF)

**S37 Fig. Hierarchical clustering of all TFs.** 2,331 TFs and cofactors were identified from TFCheckpoint2.0. TFs known to be involved in the macrophage-neutrophil decision are marked to the right of the distance matrix. See the legend of S16 Fig for the plot description.
(TIFF)

**S38 Fig. Clustering of all TFs by dominant behavior.** The TFs are ordered by their dominant behavior, annotated on the left. See the legends of S16 and S37 Figs for the plot description.
(TIFF)

## Acknowledgments

The services provided by the UND Genomics Core were supported by the National Institute of General Medical Sciences of the National Institutes of Health under Award Number U54GM128729 and Award number 2P20GM104360-06A1.

## Author contributions

**Conceptualization:** Manu.

**Formal analysis:** Trevor Long.

**Funding acquisition:** Manu.

**Investigation:** Andrea Repele, Joanna Handzlik, Nimasha Samarawickrama, Trevor Long, Sunil Nooti, Veena Potluri, Yen Lee Loh, Manu.

**Methodology:** Andrea Repele, Joanna Handzlik, Nimasha Samarawickrama, Yen Lee Loh, Manu.

**Software:** Joanna Handzlik, Nimasha Samarawickrama, Yen Lee Loh, Manu.

**Supervision:** Yen Lee Loh, Manu.

**Visualization:** Joanna Handzlik, Nimasha Samarawickrama, Trevor Long, Sunil Nooti, Yen Lee Loh, Manu.

**Writing – original draft:** Joanna Handzlik, Nimasha Samarawickrama, Yen Lee Loh, Manu.

**Writing – review & editing:** Joanna Handzlik, Nimasha Samarawickrama, Trevor Long, Sunil Nooti, Yen Lee Loh, Manu.

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
