## [Decision Letter · Decision Letter 0]

4 Aug 2025

PCOMPBIOL-D-25-01103

The differentiation of myeloid progenitors is effected by cascading waves of coordinated gene expression that remodel cellular physiology in a characteristic sequence

PLOS Computational Biology

Dear Dr. Manu,

Thank you for submitting your manuscript to PLOS Computational Biology. After careful consideration, we feel that it has merit but does not fully meet PLOS Computational Biology's publication criteria as it currently stands. Therefore, we invite you to submit a revised version of the manuscript that addresses the points raised during the review process.

Please submit your revised manuscript within 60 days Oct 04 2025 11:59PM. If you will need more time than this to complete your revisions, please reply to this message or contact the journal office at ploscompbiol@plos.org. Please include the following items when submitting your revised manuscript:

We look forward to receiving your revised manuscript.

Kind regards,

Saurabh Sinha

Academic Editor

PLOS Computational Biology

Ilya Ioshikhes

Section Editor

PLOS Computational Biology

**Journal Requirements:**

3) Thank you for stating "The raw RNA-Seq sequence data will be made available on the Sequence Read Archive of the National Center for Biotechnology Information. All code will be made available on Github."  We strongly recommend all authors decide on a data sharing plan before acceptance, as the process can be lengthy and hold up publication timelines. Please note that, though access restrictions are acceptable now, your entire data will need to be made freely accessible if your manuscript is accepted for publication. This policy applies to all data except where public deposition would breach compliance with the protocol approved by your research ethics board. If you are unable to adhere to our open data policy, please kindly revise your statement to explain your reasoning and we will seek the editor's input on an exemption. Please be assured that, once you have provided your new statement, the assessment of your exemption will not hold up the peer review process.

**Reviewers' comments:**

Reviewer's Responses to Questions

Reviewer #1: In this study, the authors analyze the dynamic transcriptomic changes occurring during of granulocte (Gr) / macrophage (Mac) differentiation. They use a IL3-dependent progenitor cell line that recapitulates Gr/Mac differentiation, in which the key myeloid specification regulator PU.1 is knocked out, and an small molecule-inducible PU.1 is added to enable inducible control of differentiation. A key feature of this dataset is high temporal resolution analysis of transcriptomic changes through bulk single-cell RNA-sequencing of differentiating cells at closely-spaced time points (25 in total per time course). The authors further used non-negative matrix factorization (NNMF) methods to assign genes into distinct groups based on their temporal expression profiles. From this analysis, the authors conclude that differentiated is characterized by successive waves of transient transcriptomic changes, that initiate at different times and correspond to distinct regulatory events, such as signaling changes, metabolic changes, and finally lineage commitment.

The main result – that there are successive waves of gene expression corresponding to signaling, metabolic, and lineage-defining gene programs – is of specific interest to those granulocyte / macrophage differentiation. However, such dynamics have already been observed in previous studies in other differentiation contexts (e.g. PMID: 23467089) and perhaps not unexpected for this study. Furthermore, the insights drawn are limited by a generally lack of perturbation approaches to probe hau underlying gene networks, as well as a lack of the deeper engagement of the specific gene programs arising under different perturbations.

Major issues

- The comparison of the in vitro differentiated cells to in vivo data utlizes global transcriptome-wide metrics to measure the distance between different cell populations. However, this analysis should be complemented by analysis for the regulation of specific genes / gene programs that may be associated with progenitor or differentiated states. For instance, what macrophage-specific genes to OHT treated PUER cells express or fail to express? Similarly, are there stem/progenitor associated programs that are expressed or not prior to induction of differentiation?

- Fig. 4: How are the cutoffs for the genes to b showed for the NMFF determined? The authors select this be the top 200 genes, but at what point down the list do these weights become negligible? Relatedly, are the red lines indicative of the weighted average of the behaviors? They do not seem anywhere close to being an average of the top of 200 genes with differential behaviors.

- Related to the point above: the NMFF groupings appear to highlight temporal differences while underplaying initial baselines or fold differences. This is concerning because though many behaviors appear to show a clear, transient pulse on average, some of the gene expression changes appear to look much more gradual. (e..g this is especially prominent for Behaviors 1 and 3 for 4-OHT only treated cells, where there is sustained expression of many genes that appear to be outside the time window of the pulse as indicated by the red lines. Because of these concerns, it would be important to (1) use a complementary method to group together genes with similar expression time course, and (2) perform this analysis not only on z-score normalized traces, but on raw traces or traces where no normalization for baseline / effect size is performed.

- Generally, there is very little further analysis of transcriptomic changes in relation to specific genes or gene regualtory networks important for Gr or Mac differentiation. Fig. 7 and Fig. 8 show gene ontologies but these do not highlight or establish specific regulators or candidates.

Minor issues

- Fig. 1D: it is hard to visualize the in vitro differentiated cells because they are very sparse compared to the cells from bone-marrow.

- Fig. 2C; labeling of the earlier time points on the heatmap will be useful as a lot of the discussion focuses on early time points (~8 hrs)

- Fig. 3A-B: It would be useful to label the specific genes that are turned on or off that corresponding to known gene regulatory events in granulocyte or macrophage differentiation.

- Fig. 3E-F: These Venn diagrams are hard to interpret and my opinion do not clearly convey the author’s points.

- Fig. 5: The 3D heatmap is hard to visualize. It would be more useful to do a 2D heatmap that is annotated and also unfolds over time, see for instance ( )

- Figure legends (Fig. 2 and thereafter): It would be useful the experiment from which the data is derived, even if briefly (e.g. one sentence indicating that the data were derived from the experiment as performed in Fig. 1). It would also be useful to briefly describe the computational method used (e.g. NMF for Fig. 4)

Reviewer #2: The manuscript by Manu et al describes a high-resolution time-course analysis of gene expression during 7 days of myeloid differentiation, utilising the PUER myeloid cell line and bulk RNAseq analysis at 27 timepoints. An advantage of the PUER system is that differentiation can be rapidly triggered by addition of 4-hydroxy-tamoxifen to activate the PUER fusion protein, and differentiation can be directed down more than one pathway. A disadvantage – as acknowledged by the authors – is that the cell line is mutated, which complicates extrapolation of its behavior to primary hematopoietic cells. Nevertheless, the study has generated interesting insights into the dynamic processes involved in differentiation to neutrophil and macrophage cell fates; specifically, the identification of two sudden changes in genome-wide gene expression at around 8h and 80h, and of 10 distinct patterns (behaviors) of gene expression that capture the behavior of almost all transcripts across the two timeseries. This data will provide a valuable resource for the scientific community to further interrogate, and to integrate with existing or future relevant datasets. I have a number of specific comments, detailed below.

1. In the introduction, the authors discuss how progenitors with more than one possible cell fate might be biased toward one fate or another by expression of a key regulator/TF. PU.1 is such a key regulator, and PUER cells provide a model for using a single TF to trigger differentiation. However, the samples designated as 0 hours have been cultured in different conditions (either IL-3 or 48h of GCSF), and the authors make it clear that there are many gene expression changes during the 48h pre-treatment with GCSF, and that the two 0h datapoints differ markedly in their gene expression (e.g. the PCA in Fig 2C, Fig 3C and also shown by Behavior 1 in fig 4; Fig S6A). In line 56 of the introduction, the authors state that GCSF can initiate fate transformation in PUER cells. It is therefore unclear to me whether the authors think that the early gene expression changes they measure are an immediate cause/consequence of lineage choice of bipotent PUER cells, or if they are capturing the early stages of differentiation triggered by PUER activation in the contexts of macrophage- and neutrophil-biased (or indeed committed) cells. In the author summary, they refer only to cell “maturation”, but elsewhere refer several times to lineage decisions: Line 384 - ..”may indicate important moments when crucial lineage decisions are made”; line 413 “…that the cell fate decision is made very early during the differentiation process”; line 431 “…that the cell-fate decision is made within the first 4 hours of OHT induction”. And in the discussion - line 569 “…that the cell-fate decision appears to have been made by the time of the 8 – 12h transition”. I agree with this final statement, but it is not clear to me that the timeseries has captured gene expression data from cells before or during the point that the cell fate decision was made. This could easily be explored by testing the lineage potential of the PUER cells pre-treated with GCSF for 48h, by switching them back into IL-3 when (or before) 4-OHT is added. Cell potential could be assessed by morphology of the differentiated cells, and/or by gene expression analysis of some key genes that distinguish between neutrophil and macrophage lineages. If the authors can demonstrate that most of the GCSF pre-treated cells have retained macrophage potential, this would indicate they have indeed analysed the neutrophil/macrophage lineage decision point, which would strengthen the paper. If the authors do not wish to address this experimentally, they should clarify the evidence that the PUER cells remain bipotent in both conditions at the timepoint they designate 0h, or re-assess their use of terms referencing lineage decisions.

2. Related to the point above, if the authors do undertake biological experiments to address cell potential, it would be interesting to know to what extent the differentiation and associated gene expression changes are dependent on continuous activation of PUER (by the addition of fresh OHT at 40h, 88h and 136h), or whether OHT can be withdrawn after the 8h or 80h transition points.

3. Although the manuscript primarily concerns genome-wide changes in the transcriptome and how these relate to cell state, the changes are being triggered by experimental manipulation of a single transcription factor. The authors discuss the possibility of conducting “multi-omic” analyses in the future to clarify the causality of events (line 574). It is surprising to me that the authors have not utilised pre-existing ChipSeq datasets for PU.1 and/or PUER in hematopoietic cells to provide some insight into which – or what proportion of - the differentially expressed genes are likely to be directly bound and regulated by PUER, and whether these are concentrated in particular gene expression behaviors. For example, genes displaying Behavior 8 (upregulated between 0h and 80h in both GCSF and IL3) seem likely to be highly enriched for direct targets of PUER. Although there are obvious reservations to using binding or chromatin accessibility data taken from different cell types/models, this should be explored.

Minor points

1. The methods state 4 biological replicates were collected for each timepoint (line 625) but many of the figures show only 3 replicates (e.g. Fig S1). Presumably the methods is a typo, or one replicate from each timepoint was held back?

2. I assume from the methods that 4OHT induction of the GCSF pre-treated samples is in the continued presence of GCSF but it would be helpful to add this to the text.

3. In fig S1, although the outlier timepoints that are identified in the text (96h and 144h) can be seen in the clustering, it would be helpful to highlight them in the timeline.

4. Figures S11 and S12 show the GO term “negative regulation of cell cycle” is enriched in both Behaviors 1 and 2, although Behavior 2 also shows enrichment of positive regulation of cell cycle. Also, the genes in Behavior 2 are transiently downregulated after addition of 4OHT but then significantly upregulated throughout most of the timecourse. I am not sure why this leads the authors to suggest that GCSF pre-treatment and addition of OHT in the presence of GCSF have opposing effects on the cell cycle (line 480), and they should explain this more clearly, or remove it from the text.

Reviewer #3: Repele and Handzlik et al. acquired bulk RNA-seq time series data at 27 time points (resolution: hourly to 8-hourly, sampled over 7 days) from PUER cells driven to macrophage (IL-3 + OHT) or neutrophil (G-CSF + OHT) fates. They applied NMF to extract 10 “behaviors” that together reconstruct >36 k transcripts; and show that differentiation proceeds through cascading, pulse-like gene-expression waves with two sharp phase transitions (at ~8 h and ~80 h) that align with functional remodeling (in terms of signaling, translation/mRNA processing, metabolism, and innate-immunity genes).

Several regulators of myeloid fate are well-known (PU.1, C/EBP-alpha, Gfi1, etc.), but how the signal from a handful of factors is propagated to thousands of target genes, and how that process is coordinated in time, is largely unknown. Earlier work offered (i) end-point RNA-seq comparisons that miss transient waves, and (ii) a few lower-resolution time-series (>=12 h spacing) that hinted at “jumps” but could not measure velocity or ordering; and no study had captured the minute-to-day continuum at genome scale. In particular, the timing, order and scale of transcriptome changes during macrophage-versus-neutrophil differentiation are poorly resolved, which the authors wanted to address in this manuscript.

It is a generally well-written and straight-forward paper. The dataset, although bulk, is a unique resource. Identifying the 7–10 patterns that reconstruct transcriptome-scale expression with high fidelity is also important. These positives, however, are diminished by several major concerns about a lack of methodological innovations and rigor, some choices in experiment design, and a lack of convincing demonstration that this work truly contributes to the literature on cell-fate mapping.

Major comments.

1. The authors have dedicated major effort to justify that their choice of PUER cells reflect in vivo biology. However, the conclusions that PUER cells approach BM cells and PUER neutrophils and macrophages are closer in their transcriptomic state to neutrophils and macrophages developing in vivo, do not convincingly justify that using PUER cells can reveal the intermediate cell states and dynamic mechanisms of hematopoiesis. So, the mechanistic value of the conclusions is questionable.

2. As a follow-up of point #1, we should avoid accepting batch-effect-corrected results at their face value. Newer methods can identify what information is lost due to batch-effect corrections. The authors should consider those methods.

3. There are no notable methodological contributions. The factors were not allowed any dynamicity, which is a problem given that gene expression changes significantly during the profiled time course. The RMSE values of NMF were small, but that is likely an artifact of scaling all gene expressions between 0 and 1.

4. Follow-up of point #3, it is important to test the assumption that the non-negative factors need not be dynamic. Does the presented NMF approach work well on single-cell lineage tracing data?

5. The predictive value of this work is questionable: there were opportunities to derive predictive ODE models, which could help testing on held-out time points.

6. It is also unclear if/how the conclusions fit with prior single-cell studies of cell-fate tracing during hematopoiesis (e.g., Weinreb et al.’s data).

**Have the authors made all data and (if applicable) computational code underlying the findings in their manuscript fully available?**

Reviewer #1: Yes

Reviewer #2: Yes

Reviewer #3: **No:** They mentioned to release data and code upon publication: "The raw RNA-Seq sequence data will be made available on the Sequence Read Archive of the National Center for Biotechnology Information. All code will be made available on Github."

PLOS authors have the option to publish the peer review history of their article (what does this mean?). If published, this will include your full peer review and any attached files.

Reviewer #1: No

Reviewer #2: No

Reviewer #3: No

**Figure resubmission:**
---

## [Decision Letter · Decision Letter 1]

29 Mar 2026

PCOMPBIOL-D-25-01103R1

The differentiation of myeloid progenitors is effected by cascading waves of coordinated gene expression that remodel cellular physiology in a characteristic sequence

PLOS Computational Biology

Dear Dr. Manu,

Thank you for submitting your manuscript to PLOS Computational Biology. After careful consideration, we feel that it has merit but does not fully meet PLOS Computational Biology's publication criteria as it currently stands. Therefore, we invite you to submit a revised version of the manuscript that addresses the points raised during the review process.

We look forward to receiving your revised manuscript.

Kind regards,

Saurabh Sinha

Academic Editor

PLOS Computational Biology

Ilya Ioshikhes

Section Editor

PLOS Computational Biology

**Journal Requirements:**

1) When completing the data availability statement of the submission form, you indicated that you will make your data available on acceptance. We strongly recommend all authors decide on a data sharing plan before acceptance, as the process can be lengthy and hold up publication timelines. Please note that, though access restrictions are acceptable now, your entire data will need to be made freely accessible if your manuscript is accepted for publication. This policy applies to all data except where public deposition would breach compliance with the protocol approved by your research ethics board.

**Reviewers' comments:**

Reviewer's Responses to Questions

Reviewer #2: Point 1 – re whether G-CSF pre-treated cells remain bi-potential and therefore the activation of PU-ER by OHT triggers neutrophil commitment of a bi-potential cell (as distinct from modelling the earliest stages of neutrophil differentiation, post-commitment). The authors have performed the suggested “switch” experiment, and I am satisfied that this demonstrates that after pre-treatment with G-CSF, at least some of the cells remain able to respond to IL-3 and are therefore not irreversibly committed to neutrophil differentiation. However, as Fig S12 presents the 96h data in the form of cell percentages, it remains a formal possibility that the monocytic cells observed after 96h in the switch condition could originate from only a small proportion of the cells pre-treated with G-CSF (which have retained bi-lineage potential), with the majority of the cells in fact having undergone some degree of neutrophil commitment. Any such neutrophil-committed cells would presumably be present in the RNA samples harvested at the time of OHT addition in the G-CSF time-course. This possibility can be readily excluded by the authors simply confirming in the text or the figure legend that the cell numbers were similar throughout the IL-3 and switch experiments. If the authors did not record cell numbers (or % viability?) in the switching experiment and so are unable to exclude this possible caveat, they should consider softening some of their statements. For example, to reconsider the statement (line 220) that “the effects of G-CSF pre-treatment are completely reversible….” . It seems a little unlikely that the large effect of G-CSF pre-treatment on gene expression (Fig 2A; -48h vs 0h) does not include an impact on some genes involved in lineage choice. I do not think it is particularly problematic if it remains somewhat open to debate whether or not the gene expression analysis of G-CSF differentiation captures the moment of lineage commitment, but is important that this is clarified.

As an additional point, fig S12E has error bars but the legend does not indicate the number of replicate experiments performed.

Point 2 – I acknowledge the authors’ opinion that this is beyond the scope of the current study.

Point 3 – The request at first review was to use published PU1 binding data to gain insight into which – or what proportion of - the differentially expressed genes are likely to be directly bound and regulated by PUER, and whether these genes are concentrated in particular gene expression behaviors.

The authors have addressed this by generating their own PU1 binding data from their differentiation model, which is preferable to using other PU1-binding data from other cell systems. They generate PU1 binding data from undifferentiated cells (presumably in IL3) and from cells after 48h of differentiation with IL3-OHT, identifying 6,724 differentially bound regions. They do not identify any of the bound genes, or comment on their identity or function, but they explore the relationship between the regions bound by PU.1 at 48h and the various patterns of differential gene expression observed during both the IL3 and G-CSF differentiations. The data is presented in the new Figures S20 and S21.

Fig S20 : As the authors say, it is clear that the gene set that is upregulated in the first hour of differentiation is enriched for genes they identify as bound by PU1 at 48h (either close to the TSS or more distally), and it is reasonable to assume that much if not all of this upregulation is due to rapid PU1 binding after its activation by OHT. I am surprised that this figure only contains the DEGs from 0h vs 1h to represent early changes in gene expression, and wonder if I am missing something? In Fig 6A, it is apparent that in the IL3 condition, relatively few DEGs are detected in 0h vs 1h, compared with 1h vs 2h, 2h vs 3h, 4h vs 8h, and even 8h vs 12h. Given this, the use in Fig S20 of only the 0h vs 1h DEGs from within the first 12 hours of differentiation seems odd, especially as (i) the 0h vs 1h DEGs have not been included in Figs 6B and 6C, and (ii) the authors have identified that a key transition in gene expression occurs at 8 hours. The most interesting insights might come from those genes that are differential between 8h and either 4h or 12h (or both). Since the authors already have the GE and the PU1 binding data, one or more of the other early timepoint comparisons should be added to Fig S20.

Fig S21 : Here the authors look for enrichment of PU1-bound genes (defined as genes/loci bound by PU1/ER after 48h of culture in IL3 + OHT) within the 10 patterns of gene expression( Behaviors 1 – 10) defined in Figure 4. This approach gives greater insight into the likely role of PU.1 in initiating and supporting differentiation, as it makes more use of the dynamics of gene expression across the 168 hours. The most marked enrichments of PUER binding are shown to be within 1 kb of the TSS of genes with expression Behaviors 2,3, 4 and 10. In the description of the results, the authors need to take more account of the fact that the binding data is derived from a “snapshot” of cells differentiated for 48 hours in IL-3, and acknowledge that it may not be correct to infer that PU1 binding around a given set of genes would be the same in neutrophil conditions as in IL3. For example, they use the enrichment of binding to genes with Behaviors 4 and 10 to support the statement (line 523) that “PU1 tends to regulate genes..…...turned on permanently at the end”. For Behavior 10 genes, this is a reasonable interpretation, as they are upregulated after ~ 60h in IL-3, and enrichment of PU1 binding is seen at 48h in IL-3. The lack of upregulation of these genes in G-CSF could be attributed to a lack of PU.1 binding or to other regulatory factors. But the situation for Behavior 4 genes is more complicated, as they are not upregulated in IL-3 – even though this is where the increase in PU1 binding was seen – but are upregulated in G-CSF, where PU1 binding can only be inferred. Inferring the contribution of PU1 binding to changes in gene expression is not a problem, as long as the authors address more specifically how strongly each inference they draw is supported by the experimental data.

Reviewer #3: The authors have included significant reanalyses and responded well about the limitations of the study.

Our point about "The factors were not allowed any dynamicity" was probably unclear to the authors. We meant that the H_t factors are not functions of time (i.e., H(t)) by design. The H_t factors are fit independently for each timepoint, there is no conditional dependency of H_{t+1} on H_t. This should be clarified in their Methods and discussed as future works.

**Have the authors made all data and (if applicable) computational code underlying the findings in their manuscript fully available?**

Reviewer #2: Yes

Reviewer #3: **No:** Not yet, but the authors committed to make data and codes public.

PLOS authors have the option to publish the peer review history of their article (what does this mean?). If published, this will include your full peer review and any attached files.

Reviewer #2: No

Reviewer #3: No

**Figure resubmission:**
---

## [Editor Report · Decision Letter 2]

27 Apr 2026

Dear Dr. Manu,

We are pleased to inform you that your manuscript 'The differentiation of myeloid progenitors is effected by cascading waves of coordinated gene expression that remodel cellular physiology in a characteristic sequence' has been provisionally accepted for publication in PLOS Computational Biology.

Best regards,

Saurabh Sinha

Academic Editor

PLOS Computational Biology

Ilya Ioshikhes

Section Editor

PLOS Computational Biology

---

## [Editor Report · Acceptance letter]

PCOMPBIOL-D-25-01103R2

The differentiation of myeloid progenitors is effected by cascading waves of coordinated gene expression that remodel cellular physiology in a characteristic sequence

Dear Dr Manu,

I am pleased to inform you that your manuscript has been formally accepted for publication in PLOS Computational Biology. Your manuscript is now with our production department and you will be notified of the publication date in due course.

With kind regards,

Anita Estes
